# Functional characterization of a 'plant-like' HYL1 homolog in the cnidarian *Nematostella vectensis* indicates a conserved involvement in microRNA biogenesis

**Abhinandan M Tripathi†, Yael Admoni†, Arie Fridrich‡, Magda Lewandowska‡, Joachim M Surm‡, Reuven Aharoni, Yehu Moran***

Department of Ecology, Evolution and Behavior, Alexander Silberman Institute of Life Sciences, Faculty of Science, The Hebrew University of Jerusalem, Jerusalem, Israel

**\*For correspondence:**
yehu.moran@mail.huji.ac.il

†These authors contributed equally to this work
‡These authors also contributed equally to this work

**Competing interest:** The authors declare that no competing interests exist.

**Abstract** While the biogenesis of microRNAs (miRNAs) in both animals and plants depends on the RNase III Dicer, its partner proteins are considered distinct for each kingdom. Nevertheless, recent discovery of homologs of Hyponastic Leaves1 (HYL1), a 'plant-specific' Dicer partner, in the metazoan phylum Cnidaria, challenges the view that miRNAs evolved convergently in animals and plants. Here, we show that the HYL1 homolog Hyl1-like a (Hyl1La) is crucial for development and miRNA biogenesis in the cnidarian model *Nematostella vectensis*. Inhibition of Hyl1La by morpholinos resulted in metamorphosis arrest in *Nematostella* embryos and a significant reduction in levels of most miRNAs. Further, meta-analysis of morphants of miRNA biogenesis components, like Dicer1, shows clustering of their miRNA profiles with Hyl1La morphants. Strikingly, immunoprecipitation of Hyl1La followed by quantitative PCR revealed that in contrast to the plant HYL1, Hyl1La interacts only with precursor miRNAs and not with primary miRNAs. This was complemented by an in vitro binding assay of Hyl1La to synthetic precursor miRNA. Altogether, these results suggest that the last common ancestor of animals and plants carried a HYL1 homolog that took essential part in miRNA biogenesis and indicate early emergence of the miRNA system before plants and animals separated.

## Editor's evaluation

This paper will be of importance for researchers in the field of RNA biology and evolutionary biology. It provides a new perspective on the origins of the miRNA pathways, and proposes a common origin of plant and animal miRNA pathways. The main conclusions of the paper are well supported.

## Introduction

MicroRNAs (miRNAs) are 21–24 nucleotides-long small RNAs that are known to be involved in post-transcriptional gene regulation and play important roles in both plant and animal development (*Alvarez-Garcia and Miska, 2005*; *Bråte et al., 2018*; *Voinnet, 2009*). The miRNA is transcribed by RNA polymerase II into a long primary transcript, which is further processed into a miRNA precursor and finally chopped into ~22 nucleotide miRNA/miRNA* duplex (*Bartel, 2004*; *Bartel, 2018*; *Voinnet, 2009*). The processing of miRNA varies between plants and animals (*Moran et al., 2017*). In animals,

**eLife digest** In both animals and plants, small molecules known as micro ribonucleic acids (or miRNAs for short) control the amount of proteins cells make from instructions encoded in their DNA. Cells make mature miRNA molecules by cutting and modifying newly-made RNA molecules in two stages.

Some of the components animals and plants utilize to make and use miRNAs are similar, but most are completely different. For example, in plants an enzyme known as Dicer cuts newly made RNAs into mature miRNAs with the help of a protein called HYL1, whereas humans and other animals do not have HYL1 and Dicer works with alternative partner proteins, instead. Therefore, it is generally believed that miRNAs evolved separately in animals and plants after they split from a common ancestor around 1.6 billion years ago.

Recent studies on sea anemones and other primitive animals challenge this idea. Proteins similar to HYL1 in plants have been discovered in sea anemones and sponges, and sea anemone miRNAs show several similarities to plant miRNAs including their mode of action. However, it is not clear whether these HYL1-like proteins work in the same way as their plant counterparts.

Here, Tripathi, Admoni et al. investigated the role of the HYL1-like protein in sea anemones. The experiments found that this protein was essential for the sea anemones to make miRNAs and to grow and develop properly. Unlike HYL1 in plants – which is involved in both stages of processing newly-made miRNAs into mature miRNAs – the sea anemone HYL1-like protein only helped in the second stage to make mature miRNAs from intermediate molecules known as precursor miRNAs.

These findings demonstrate that some of the components plants use to make miRNAs also perform similar roles in sea anemones. This suggests that the miRNA system evolved before the ancestors of plants and animals separated from each other. Questions for future studies will include investigating how plants and animals evolved different miRNA machinery, and why sponges and jellyfish have HYL1-like proteins, whereas humans and other more complex animals do not.

the biogenesis of miRNAs is compartmentalized as the processing occurs in both the nucleus and cytoplasm. Within the nucleus, the RNase type III Drosha and its partner Pasha (also called DGCR8) constitute a microprocessor complex (*Han et al., 2004b*; *Kim et al., 2009*). This complex acts on primary miRNA (pri-miRNA) transcripts and processes them into precursor miRNA (pre-miRNA). The pre-miRNA is then transported by Exportin 5 into the cytoplasm where they get further processed into the mature miRNA by the RNase type III Dicer with the help of other double-stranded RNA binding proteins such as loquacious (Loqs), transactivation response element RNA-binding protein (TRBP), and protein activator of the interferon-induced protein kinase (PACT) (*Han et al., 2004b*; *Redfern et al., 2013*; *Saito et al., 2005*). Contrastingly, in plants both pri-miRNA and pre-miRNA are processed into mature miRNA by a single RNase type III, called DICER-LIKE1 (DCL1) assisted by its partner the double-stranded RNA-binding motif (DSRM)-containing protein, Hyponastic Leaves1 (HYL1) within the nucleus (*Han et al., 2004a*; *Voinnet, 2009*). In both plants and animals, the mature miRNA duplex interacts with Argonaute proteins (AGOs) and forms the RNA-induced silencing complex (RISC) in the cytoplasm. The RISC commences miRNA guided cleavage or translational inhibition of complementary targets genes (*Kim et al., 2009*).

The metazoan lineages of Bilateria and its sister group Cnidaria separated more than 600 million years ago. While Bilateria include the vast majority of animals, Cnidaria include sea anemones, corals, hydroids, and jellyfish. The phylogenetic position of cnidarians makes them an important comparative group for inferring animal evolution. In a previous study, we identified different components of miRNA biogenesis machinery in Cnidaria and observed that most bilaterian components have cnidarian homologs. However, cnidarians lack homologs of classical bilaterian Dicer protein partners such as PACT, Loqs, or TRBP (*Moran et al., 2013*). Interestingly two homologs of HYL1 called Hyl1-Like-a (NveHyl1La) and Hyl1-Like-b (NveHyl1Lb) were identified in the sea anemone *Nematostella vectensis* (*Moran et al., 2013*). Apart from this, it was also found that cnidarian miRNAs possess several interesting features that are common to their counterparts in plants: cnidarian miRNAs and their targets show perfect complementarity and frequently regulate their targets by messenger RNA (mRNA) cleavage (*Moran et al., 2014*). Recently, some other common features with plants were identified in

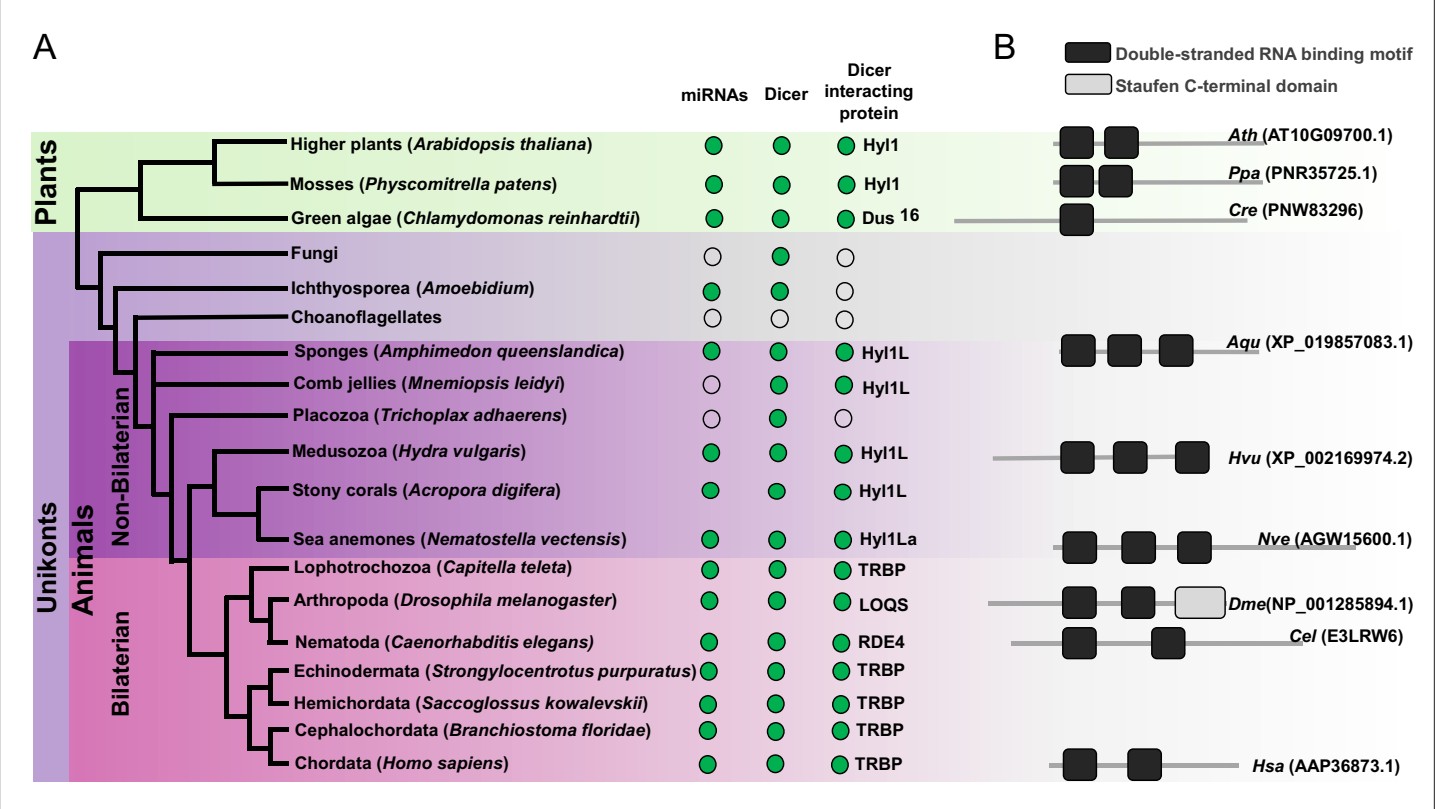

**Figure 1.** Schematic representation of a phylogenetic tree of Eukaryotes at the phylum level. (**A**) Phylogenetic tree representing the presence (green circles) and absence (open circles) of microRNAs (miRNAs), Dicer, and Dicer interacting proteins in different plant and animal phyla. The names of representatives of different phyla are given in brackets. The names of Dicer interacting proteins are given near the green circles. (**B**) Domain structure of different Dicer interacting proteins predicted by using the Pfam (https://pfam.xfam.org/). NCBI gene ID is shown in brackets.

*Nematostella*, including methylation of miRNAs by HEN1 (**Modepalli et al., 2018**), a feature rarely found in animals and the origin of miRNAs from inverted duplication of their target genes (**Fridrich et al., 2020**), a feature previously considered specific to plant miRNAs.

In addition to the presence of HYL1 homologs in Cnidaria, homologs are also present in other non-bilaterian animals such as sponges (*Amphimedon queenslandica*) and in ctenophores (*Mnemiopsis leidyi*) (**Figure 1**; **Moran et al., 2013**). However, we could not detect HYL1 homologs in Placozoa (*Trichoplax adhaerens*) (**Figure 1A**). Additionally, we also could not find any homologs in bilaterian animals and in unicellular organism like Fungi and Ichthyosporea. However, deep phylogenetic study of DSRM proteins showed that those of protozoans and fungi are phylogenetically closer to the DSRM proteins of plants (**Dias et al., 2017**). These results suggested that the HYL1-like proteins were already present in the common ancestor of plants and animals and during evolution have been lost in Bilateria and Ichthyosporea. These sequence-based observations led us to experimentally test the function of a HYL1 homolog of *Nematostella,* which could provide better insight into the evolution and origin of the miRNA biogenesis pathway. Our results show that Hyl1La is essential for the development of *Nematostella* and miRNA biogenesis, suggesting a common evolutionary history of miRNA biogenesis in plants and animals.

## Results
### Hyl1La plays an essential role in *Nematostella* development

Mutants of miRNA biogenesis pathway components exhibit severe developmental defects in both plants and animals (**Alvarez-Garcia and Miska, 2005**; **Schauer et al., 2002**). The HYL1 protein is known to play an essential role in the growth and development of the model plant *Arabidopsis thaliana* by regulating miRNA biogenesis (**Achkar et al., 2018**; **Han et al., 2004a**). Similarly, in mice,

TRBP mutants show multiple developmental abnormalities and a reduction in miRNA accumulation (*Koscianska et al., 2011*; *Zhong et al., 1999*). The *Hyl1La* gene of *Nematostella* contains 11 exons and 10 introns predicted to code for a protein containing three DSRM domains (*Figures 1B and 2A*). Unlike its paralog *Hyl1Lb* that is specific to stinging cells and carries additional protein domains, *Hyl1La* expression is ubiquitously distributed throughout *Nematostella* tissues and shares its domain structure with other cnidarian HYL1 homologs (*Moran et al., 2013*). Thus, we decided to focus our analysis on this gene. To decipher the function of Hyl1La in *Nematostella*, we designed two different splicing morpholinos (MOs) (Hyl1La SI MO1 and Hyl1La SI MO2) to knockdown by mis-splicing the gene at two different intron-exons junctions. Additionally, the gene was also targeted for inhibition by using a translation-blocking MO (Hyl1La TB MO) which binds on the 5′ UTR and sterically blocks translation (*Figure 2A*). We injected each of the three MOs into *Nematostella* zygotes in parallel with a control MO designed to bind to no target in the sea anemone genome. The effect of SI MOs was validated by PCR followed by cloning and sequencing which revealed intron retention in both cases (*Figure 2—figure supplement 1—source data 1*, *Supplementary file 1*). All the injected animals were studied until 9 days post-fertilization (dpf). We observed that more than 80% of the animals injected with control MO developed normally and metamorphosed into primary polyps. In contrast, the animals injected with any of the three Hyl1La MOs showed developmental abnormalities, where more than 90% of the animals did not settle and metamorphosed into primary polyps until 9 dpf (*Figure 2B–D and F*). The developmental abnormalities observed here were grossly similar to those observed in *Nematostella* morphants of other miRNA processing components such as HEN1, Dicer1, AGO1, and AGO2 knockdown animals (*Fridrich et al., 2020*; *Modepalli et al., 2018*; *Figure 2E*). These results indicate that Hyl1La plays an essential role in *Nematostella* development, possibly by regulating the processing and expression of miRNAs.

## Hyl1La regulates the miRNA biogenesis

The above observed metamorphosis arrest suggested the possible involvement of Hyl1La in miRNA biogenesis, as mutants defective in their miRNA biogenesis exhibit abnormal development in both animals and plants (*Achkar et al., 2018*; *Zhong et al., 1999*). HYL1 in *Arabidopsis* interacts with the stem region of miRNA precursors by using its DSRM domains and works with DCL1 synergistically (*Song et al., 2007*; *Wu et al., 2007*). Although the Dicer homolog alone is capable of processing the precursor into a mature miRNA, the presence of HYL1 is essential as it enhances the efficiency and in some cases the accuracy of miRNA biogenesis in plants (*Dong et al., 2008*; *Szarzynska et al., 2009*). To assay the possible role of Hyl1La on miRNA expression in *Nematostella*, we performed small RNA sequencing of animals, injected with Hyl1La SI MO1 and with control MO. The analysis of read length distribution showed that the small RNA reads that lied between the size of miRNAs (20–24 nt) were higher (p < 0.01, Student's t-test) in control as compared to knockdown embryos (*Figure 3A*). Further, we analyzed the miRNA expression by using miRProf (*Stocks et al., 2012*) and normalized the miRNA reads in transcripts per million (TPM) (*Supplementary file 2*). For the miRNA quantification we used the most recent *Nematostella* miRNA datasets that were obtained by AGO immunoprecipitation (IP) (*Fridrich et al., 2020*). The expression of normalized miRNA reads was compared between control and Hyl1La SI MO1. About 54% of the total identified miRNAs showed downregulation of more than twofold in Hyl1La SI MO1-injected animals as compared to the control (*Figure 3B*). Further, a significant reduction in overall miRNA abundance was observed in the knockdown morphants (p < 0.00001, Wilcoxon signed-rank test) (*Figure 3C*). The expression variation caused by the action of other two MOs (Hyl1La SI MO2 and Hyl1La TB) was also assayed by quantitative stem-loop PCR of five miRNAs (*Figure 3D* and *Figure 3—figure supplement 1*). Significant downregulation of three miRNAs: miR2022-3p, miR2025-3p, and miR2026-5p was detected in all three MOs (with the exception of miR-2026–5p with SI MO1), which supported the small RNA sequencing results. In contrast, two miRNAs, miR2027-5p and miR2028-5p, either showed upregulation or were not significantly affected by the Hyl1La knockdown. Previous studies have also shown that these two miRNAs may respond differently to other miRNAs in HEN1 and Dicer1 knockdown morphants of *Nematostella* (*Modepalli et al., 2018*). This might be relevant to other miRNAs that did not show downregulation and suggest differences in their biogenesis. Further, we also checked for the processing accuracy of all the identified miRNAs by mapping them onto their respective precursors. The analysis did not reveal any aberrant processing. These results suggest that like its

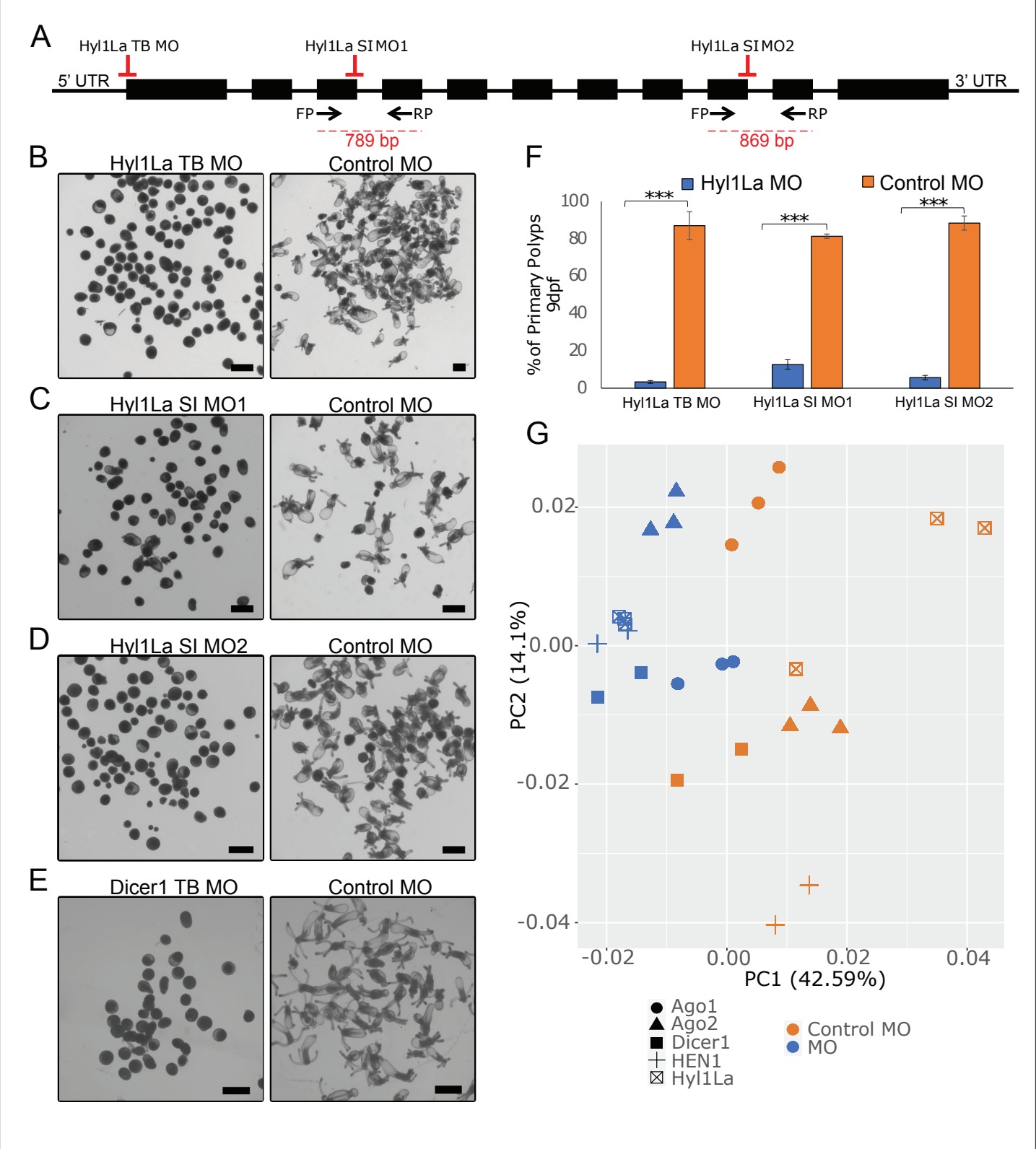

**Figure 2.** Developmental defects in different morphants of Hyl1-like a (Hyl1La). (**A**) Schematic representation of the Hyl1La gene showing the intron-exon junction as defined by comparing the transcript (NCBI Accession KF192067.1) to the *Nematostella vectensis* genome. The positions targeted by different morpholinos used in the study are shown by red symbols. The black arrows represent the position of primers designed for the validation of splicing morpholino and the product size is indicated below. (**B–D**) Images of 9 days post-fertilization (dpf) animals showing similar developmental

*Figure 2 continued on next page*

*Figure 2 continued*

defects in different morphants. (**E**) Images of 10 dpf Dicer1 morphants showing similar developmental defects to Hyl1La morphants. Scale bars are 500 µm. (**F**) Bar chart representing percentage of developed and undeveloped animals for each of the morphants. More than 80% of Hyl1La-depleted animals did not develop into the primary polyp stage after 9 dpf. Data was taken in triplicates, in each n = 200, ***p < 0.001 (Student's t-test). (**G**) Principal component analysis (PCA) plot of the miRNA expression following the knockdown of miRNA biogenesis components: Hyl1La, HEN1, Dicer2, AGO1, and AGO2. Morphants are in blue and control in orange, different symbols represent different miRNA biogenesis components and their respective controls from the same experiment.

The online version of this article includes the following source data and figure supplement(s) for figure 2:

**Figure supplement 1.** Gel image showing aberrant splicing of Hyl1-like a (Hyl1La).

**Figure supplement 1—source data 1.** Related to *Figure 2—figure supplement 1* – This data includes the gel image of aberrant Hyl1-like a (Hyl1La) splicing after different morpholinos injections.

homolog in plants, Hyl1La in *Nematostella* might be involved in enhancement of Dicer efficiency and is not involved in size selection.

To further explore the effect Hyl1La knockdown has on the expression of mature miRNAs in *Nematostella*, we performed a meta-analysis comparing the effect of knocking down previously characterized

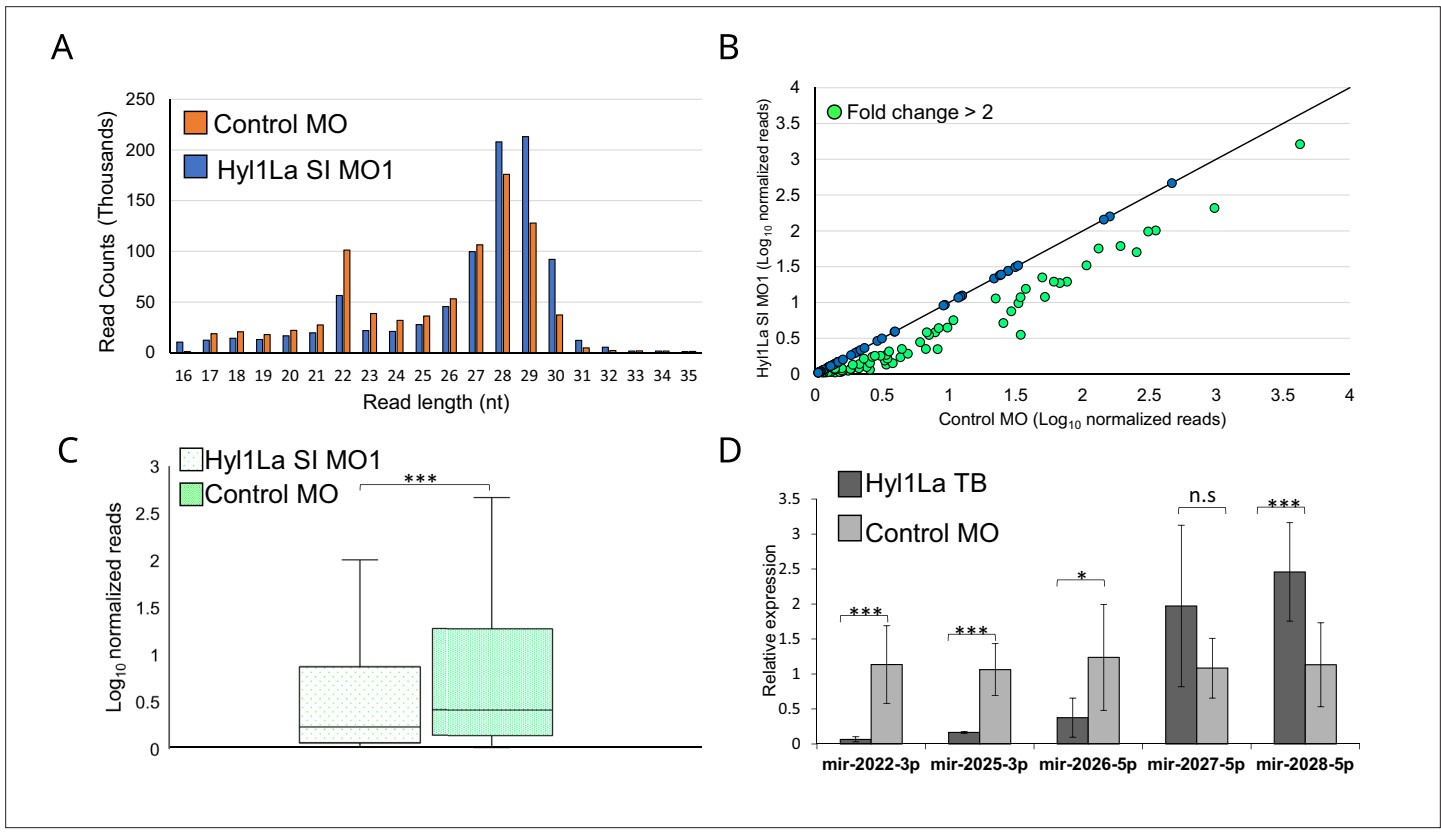

**Figure 3.** Hyl1-like a (Hyl1La) morphants show reduced expression of microRNAs (miRNAs). (**A**) Average read length distribution of small RNA reads after adapter removal. (**B**) Scatter plot representing normalized read counts of miRNAs in control and treated animals. Each dot represents the average expression of an individual miRNA. The miRNAs showing a depletion greater than twofold are indicated in green. The axes are scaled to Log10 of normalized read counts. The data represents the mean of three independent biological replicates. (**C**) Box plot showing the average abundance of miRNA read counts in Hyl1La SI MO1 and control MO. A significant reduction of miRNA read counts is noted in Hyl1La SI MO1 (p < 0.0001, Wilcoxon signed-rank test). The data represents the mean of three independent biological replicates ± SD. (**D**) Bar plot showing the expression of miR-2022, miR-2025, miR-2026, miR-2027, and miR-2028 as quantified using stem-loop PCR in translation-blocking (TB) and control morpholino. The data represents the mean of three independent biological replicates ± SD. ***p < 0.001, **p ≤ 0.01, *p ≤ 0.05 (Student's t-test), n.s. (not significant).

The online version of this article includes the following figure supplement(s) for figure 3:

**Figure supplement 1.** Effect of Hyl1-like a (Hyl1La) depletion on microRNAs (miRNA) expression.

**Figure supplement 2.** Effect of Hyl1-like a (Hyl1La) depletion on siRNA and piRNA expression.

miRNA biogenesis components such as HEN1, Dicer1, AGO1, and AGO2 (Fridrich et al., 2020; *Modepalli et al., 2018*; *Figure 2G* and *Supplementary file 3*). Using principal component analysis (PCA), we see that broadly the expression of mature miRNAs following the knockdown of Hyl1La behaves similarly to other characterized miRNA processing components. This is evident with Hyl1La morphants clustering with all miRNA biogenesis components compared to control, particularly with HEN1 and Dicer1. We also observe that both AGO1 and AGO2 are the most distant among the miRNA biogenesis components, which is consistent with their function of loading and therefore affecting a particular subset of miRNAs (*Fridrich et al., 2020*), while Hyl1La, HEN1, and Dicer1 affect the majority of the same miRNA in the same manner.

Next, we explored the effect of Hyl1La MO on other categories of small RNAs (siRNA and piRNA) reported in *Nematostella* (*Modepalli et al., 2018*; *Praher et al., 2017*). The expression of normalized reads of siRNA and piRNAs were compared between control and Hyl1La MO-injected animals. Significant ($p < 0.00001$, Wilcoxon signed-rank test) down- and upregulation of siRNAs and piRNAs, respectively, were observed in Hyl1La MO-depleted animals (*Figure 3—figure supplement 2A-D*). In case of siRNAs the difference was very mild as only 9% (46 out of 469) of siRNAs showed downregulation of more than twofold in Hyl1La MO-injected animals (*Figure 3—figure supplement 2A*). Contrastingly, most of the piRNAs showed upregulation in Hyl1La-depleted animals wherein 36% (91 out of 251) of piRNAs showed upregulation of more than twofold (*Figure 3—figure supplement 2C*). In addition, the size distribution of reads shows upregulation ($p < 0.05$, Student's t-test) of sizes that correspond to piRNAs (28–30 nt) (*Figure 3A*). Overall small RNA analysis showed that Hyl1La depletion has strong effect on miRNA expression, while mild and opposite effect on siRNA and piRNA expression, respectively.

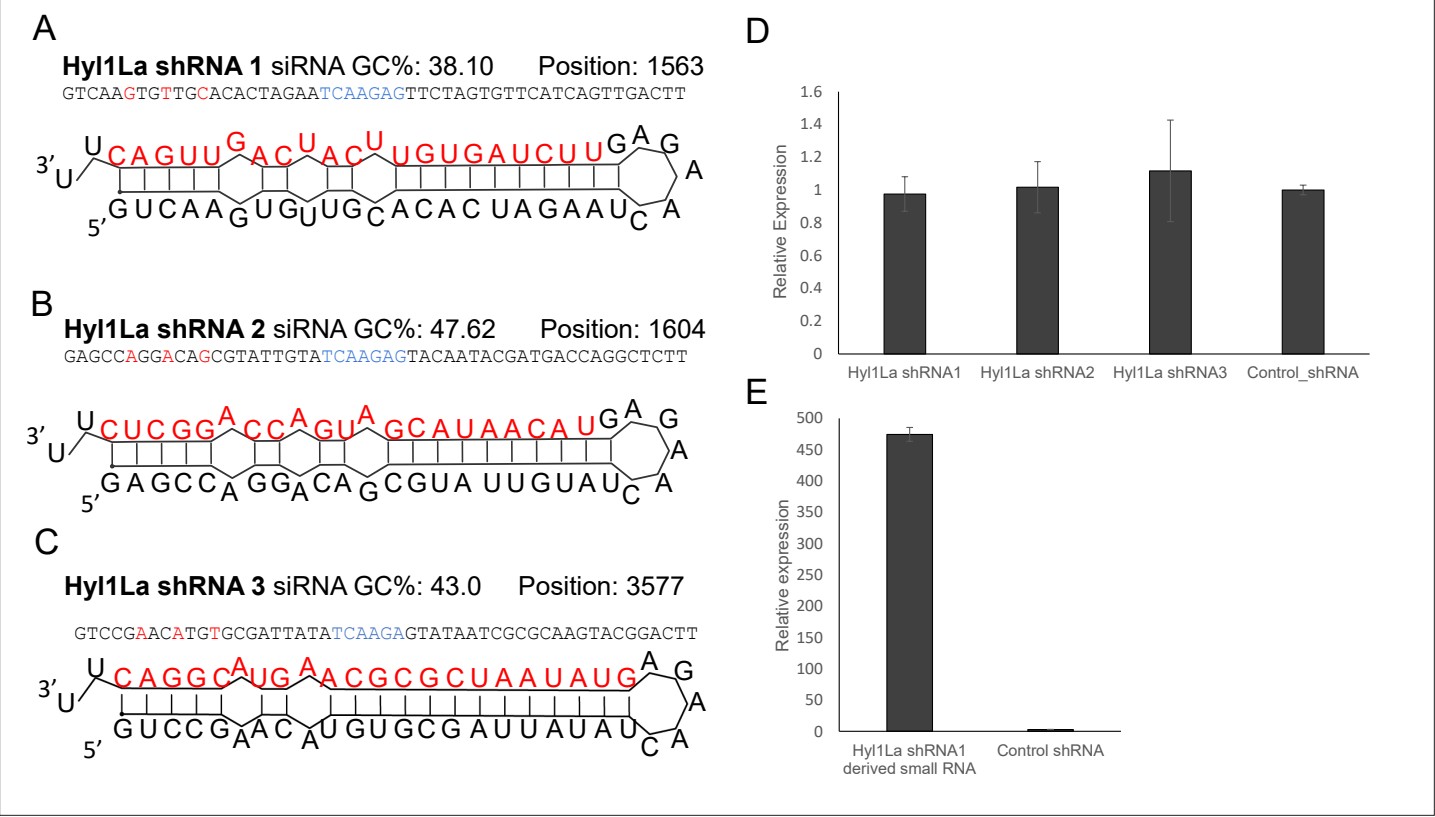

**Figure 4.** Structure of short-hairpin RNA (shRNA) precursors and their effect on Hyl1-like a (Hyl1La) expression. (**A–C**) Structure of different shRNAs designed from different positions of Hyl1La gene along with GC content and their position are shown. In the shRNA sequence, the red color shows the nucleotides edited for mismatch and blue color represents loop region. The red colored nucleotides on precursor's structure indicate the small RNA derived from the shRNAs. (**D**) Real-time quantification of Hyl1La from animals injected with different shRNAs relative to control. The data represents the mean of three independent biological replicates ± SD. (**E**) Quantification of small RNAs produced from Hyl1La shRNA1. The quantification was performed by using stem-loop qRT-PCR.

To further support our results obtained with MOs, we attempted to knock down this gene by using short-hairpin RNAs (shRNAs), a method previously established in *Nematostella* (*Karabulut et al., 2019*) that worked well in our lab for other genes (*Lewandowska et al., 2021*). We designed three different shRNAs from three different regions of Hyl1La gene (Hyl1La shRNA1, Hyl1La shRNA2, and Hyl1La shRNA3) (*Figure 4A–C*) and injected them into *Nematostella* zygotes. In parallel we also used a control shRNA with no target in *Nematostella* genome that was previously used as control for similar experiments (*He et al., 2018*; *Karabulut et al., 2019*). To assess the effect of these shRNAs on Hyl1La expression, we performed qRT-PCR from 3-day-old injected animals. Unexpectedly, we did not find any difference in Hyl1La expression (*Figure 4D*). Additionally, we also assessed the phenotype, but we could not identify any phenotypic difference either. Next, we employed stem-loop PCR to test whether small RNAs are generated from an injected shRNA and indeed the small RNAs were produced as expected (*Figure 4E*). This result indicates that the small RNAs derived from the shRNAs were not able to downregulate Hyl1La.

## Hyl1La interacts with pre-miRNAs but not with pri-miRNAs

The above observed reduction in miRNA expression indicates the possible involvement of Hyl1La in miRNA biogenesis. Being a DSRM-containing protein, it could interact with either pri-miRNA, pre-miRNA, or with both. Hence to test if Hyl1La interacts with pre- and/or pri-miRNA, we conducted an IP assay by injecting a plasmid carrying a cassette encoding an N-terminal 3 × FLAG-tagged full-length Hyl1La ('FLAG-Hyl1La') followed by a 3'-memOrange2 separated by a P2A self-cleaving peptide (*Figure 5A*; *Kim et al., 2011*; *Shaner et al., 2008*). The expression of the FLAG-Hyl1La cassette was confirmed by visualizing the animals under fluorescence microscope (*Figure 5B*) and by using anti-FLAG western blot (*Figure 5—figure supplement 1—source data 1*; *Figure 5—figure supplement 1—source data 2*). Further, RNA immunoprecipitation (RIP) with anti-FLAG antibody was performed (*Figure 5—source data 1* and *Figure 5—source data 2*) in three different biological replicates followed by qPCR analysis. Interestingly, we observed that there was very poor enrichment of pri-miRNA (Ct-values >30, sometimes undetected) as compared to pre-miRNA (*Supplementary file 4* and *Figure 5—figure supplement 1B, C*; *Figure 5—figure supplement 1—source data 3* and *Figure 5—figure supplement 1—source data 4*). Due to the very high Ct values measured for pri-miRNA, we were not able to compare the levels of pri-miRNAs bound with IgG and accurately compare them to FLAG-Hyl1La. Interestingly, when we compared the enrichment of pre-miRNAs, we found that six out of eight were significantly enriched in FLAG-Hyl1La IP in comparison to IgG, with miR-2023 showing a trend similar to other miRNAs but not reaching significance (*Figure 5D* and *Figure 5—figure supplement 1—source data 3* and *Figure 5—figure supplement 1—source data 4*). Expectedly, miR-2029 showed no enrichment in anti-FLAG samples. This miRNA might be processed independently of Hyl1La since it is also not significantly decreased following Hyl1La and Dicer1 knockdowns (*Modepalli et al., 2018*). Taken together, these results showed that Hyl1La interacts with pre-miRNA but not with pri-miRNA.

Next, we generated libraries from FLAG-Hyl1La immunoprecipitated samples and analyzed the expression of additional longer (>100 bp) transcripts (*Figure 5—figure supplement 4* and *Supplementary file 5*). We observed no significant enrichment for pri-miRNA, mRNA, non-coding RNA (rRNA, snoRNA, and tRNA) and repetitive elements. These results suggest that Hyl1La specifically binds pre-miRNAs. To further validate Hyl1La affinity to pre-miRNAs, we tested Hyl1La binding to pre-miRNAs via in vitro binding assay. We generated synthetic pre-miRNAs based on the pre-miR-2022 backbone with the mature miRNA sequence changed so it will not target *Nematostella* genes. As a control we used a shuffled version of the same pre-miRNA that does not create a hairpin secondary structure (*Figure 6A–B*). Both pre-miRNAs were labeled with biotin at their 3' end and incubated with protein lysates from FLAG-Hyl1La expressing animals in three biological replicates. Western blot conducted with anti-FLAG antibodies revealed Hyl1La binding the pre-miRNA with significantly higher efficiency than the shuffled sequence (*Figure 6C, D*; *Figure 6—source data 1*; *Figure 6—source data 2*; *Figure 6—source data 3*; *Figure 6—source data 4*; *Figure 6—source data 5*; *Figure 6—source data 6* and *Supplementary file 6*), thus supporting the specific affinity of Hyl1La to miRNA precursors.

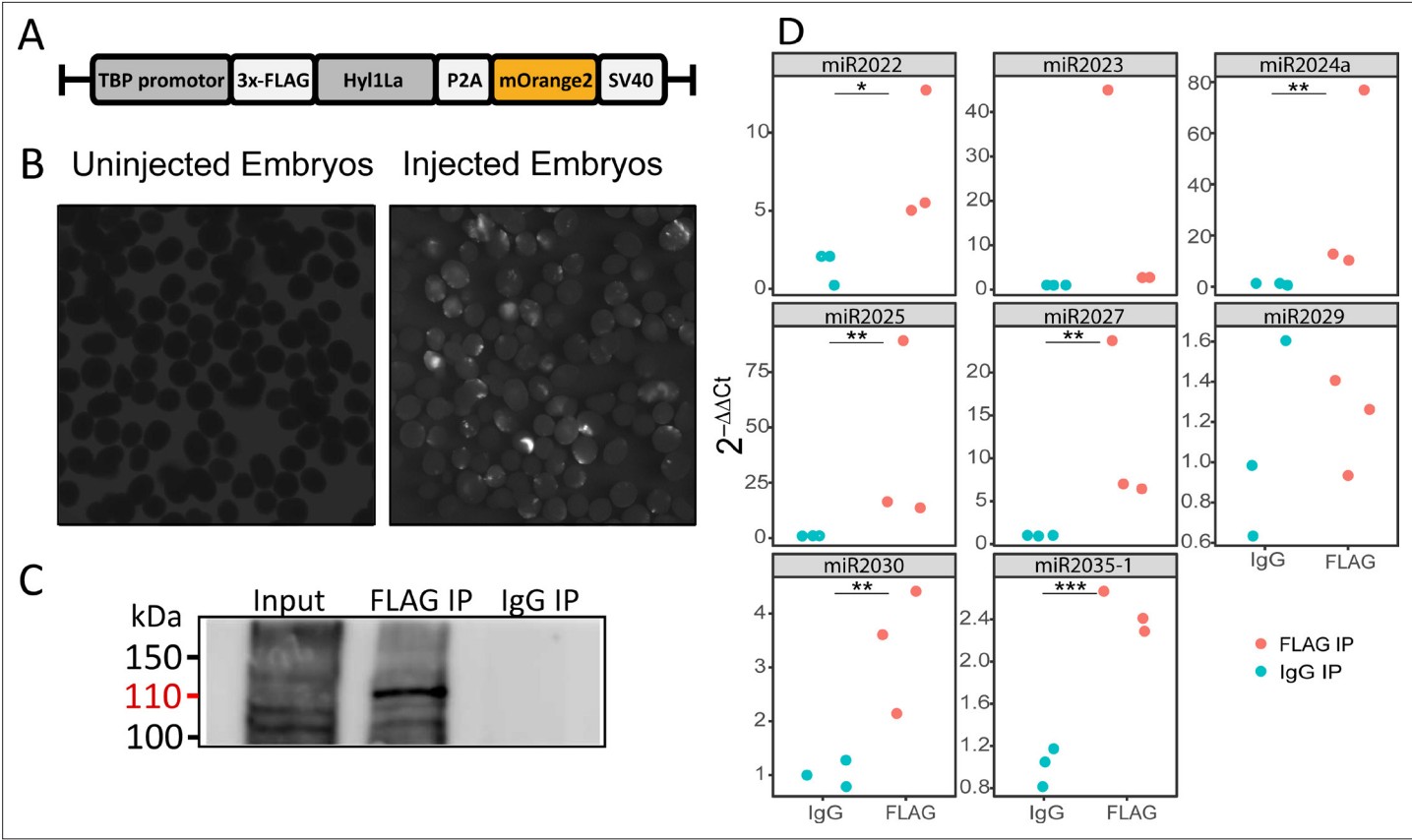

**Figure 5.** RNA immunoprecipitation (RIP) and qRT-PCR. (**A**) Schematic representation of the FLAG-Hyl1-like a (Hyl1La) construct with a TBP promoter, a self-cleaving P2A sequence, a memOrange2 gene, and the polyadenylation signal SV40. (**B**) The plasmid-injected and -uninjected embryos were visualized under a florescence microscope after 2 days. The injected embryos were showing the expression of memOrange2 (right side). (**C**) Immunoprecipitation (IP) of 3 × FLAG-Hyl1La with mouse anti-FLAG antibody or whole mouse IgG by using Protein G Magnetic Beads. The input and IP samples were subjected to Western blot with mouse anti-FLAG antibody. The red arrow (110 kDa) indicates the 3 × FLAG-Hyl1La (*Figure 5— source data 1* and *Figure 5—source data 2*). (**D**) pre-miRNA expression of eight different miRNAs were measured using the qRT-PCR. The Y-axis represents the $2^{-\Delta\Delta Ct}$ values of three independent biological replicates. \*\*\*p < 0.001, \*\*p ≤ 0.01, \*p ≤ 0.05 (Student's t-test).

The online version of this article includes the following source data and figure supplement(s) for figure 5:

**Source data 1.** Related to *Figure 5C* – Western blot of FLAG-Hyl1-like a (Hyl1La) after immunoprecipitation.

**Source data 2.** Related to *Figure 5C* – Figure includes the image of Western Blot Protein Ladder used as a size ruler.

**Figure supplement 1.** RNA immunoprecipitation (RIP) and PCR (related to *Figure 5B–D*).

**Figure supplement 1—source data 1.** Related to *Figure 5—figure supplement 1A* – Western blot of FLAG-Hyl1-like a (Hyl1La) with anti-FLAG antibody.

**Figure supplement 1—source data 2.** Related to *Figure 5—figure supplement 1A* – Figure includes the image of Western Blot Protein Ladder used as a size ruler.

**Figure supplement 1—source data 3.** Related to *Figure 5—figure supplement 1B* – Gel image of PCR amplified pri-microRNA (miRNA).

**Figure supplement 1—source data 4.** Related to Figure 5 – Figure supplement 1C – Gel image of PCR amplified pre-microRNA (miRNA).

**Figure supplement 2.** Position of primers on pre-microRNA (miRNA).

**Figure supplement 3.** Position of pre- and pri-microRNA (miRNA) primers on probable sequence of pri-miRNA.

**Figure supplement 4.** RNA-seq of RNA immunoprecipitation (RIP).

## Discussion

Altogether the absence of animal-like Dicer partner proteins such as TRBP or PACT and presence of a functional homolog of HYL1 (Hyl1La) in *Nematostella* indicated that a Hyl1-like protein might have been present in the last common ancestor of plants and animals. Apart from *Nematostella*, the presence of HYL1 homologs in additional members of Cnidaria and other non-bilaterian metazoan

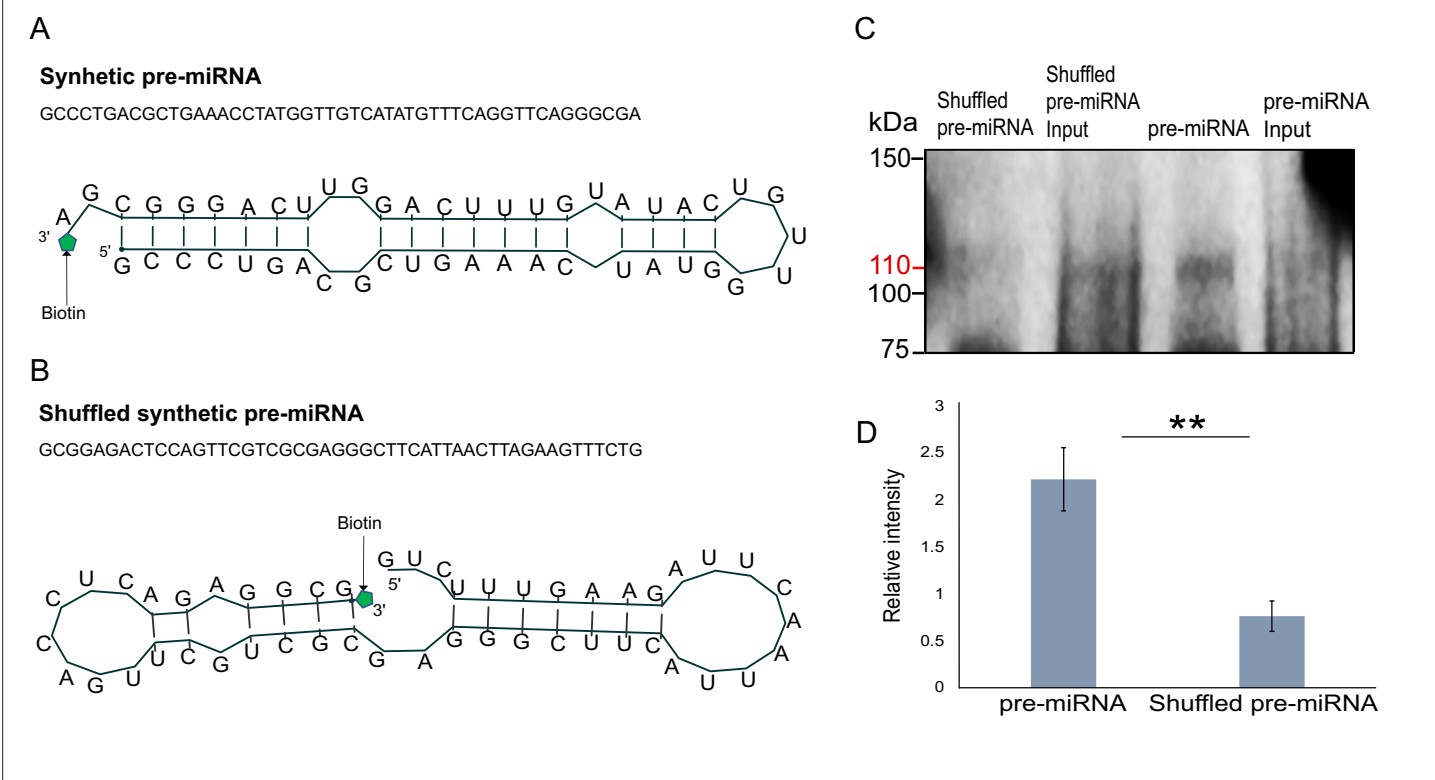

**Figure 6.** In vitro binding assay. (**A**) The sequence and secondary structure of biotin-labeled synthetic pre-microRNA (miRNA) used for in vitro binding assay. (**B**) The sequence and secondary structure of biotin-labeled shuffled synthetic pre-miRNA used as negative control for in vitro binding assay. (**C**) Pull-down of biotin-labeled synthetic pre-miRNA and shuffled pre-miRNA negative control using streptavidin magnetic beads. The pull-down samples were subjected to Western blot with mouse anti-FLAG antibody (*Figure 6—source data 1*; *Figure 6—source data 2*; *Figure 6—source data 3*; *Figure 6—source data 4*; *Figure 6—source data 5*; *Figure 6—source data 6*). (**D**) Relative intensity of Western blot bands with mouse anti-FLAG antibody, showing pull-down of biotin-labeled synthetic pre-miRNA and shuffled pre-miRNA negative control. Error bars correspond to standard deviation among replicates (n = 3). **p ≤ 0.01 (Student's t-test).

The online version of this article includes the following source data for figure 6:

**Source data 1.** Related to *Figure 6C* – Western blot of biotin pull-down with anti-FLAG antibody – replicate1.

**Source data 2.** Related to *Figure 6C* – Figure includes the image of Western Blot Protein Ladder used as a size ruler – replicate1.

**Source data 3.** Related to *Figure 6C* – Western blot of biotin pull-down with anti-FLAG antibody – replicate2.

**Source data 4.** Related to *Figure 6C* – Figure includes the image of Western Blot Protein Ladder used as a size ruler – replicate2.

**Source data 5.** Related to *Figure 6C* – Western blot of biotin pull-down with anti-FLAG antibody – replicate3.

**Source data 6.** Related to *Figure 6C* – Figure includes the image of Western Blot Protein Ladder used as a size ruler – replicate3.

groups such as sponges (*Figure 1A*) further strengthens the notion of common ancestry of the miRNA systems of plants and animals. Further, while the cleavage mode of action and nearly perfect target binding could have evolved convergently in plants and cnidarians (*Moran et al., 2014*), the involvement in miRNA biogenesis of the Hyl1L in Cnidaria and its plant homolog HYL1 is far less likely to be the result of parallel evolution. This is because the former might be driven by functional constraints or advantage in cleaving miRNA targets whereas the latter would require the independent recruitment of the same protein into the system. Thus, our results call into question the hypothesis that miRNAs evolved convergently in plants and animals from an ancestral RNAi system (*Axtell et al., 2011*; *Tarver et al., 2012*). To the best of our knowledge, our study provides the first demonstration that an elaborate miRNA biogenesis pathway might have existed in the common ancestor of plants and animals. However, it is also important to note that the presence of HYL1-like proteins by itself cannot be considered a definitive hallmark for the existence of miRNAs in non-bilaterian animals or their relatives. For example, a homolog of HYL1 exists in the ctenophore *M. leidyi*, but this species does not produce miRNAs (*Figure 1A*; *Maxwell et al., 2012*). Moreover, several unicellular relatives

of animals contain miRNAs, but no homologs of HYL1 (*Bråte et al., 2018*). Thus, this pathway might demonstrate in various lineages compositional flexibility as well as high loss rates (reviewed in *Moran et al., 2017*). Recently, it was found in *Chlamydomonas* (a unicellular green algae) that DUS16, which is a DSRM protein, and the RNase III DCL3 were efficient enough for miRNA processing (*Yamasaki et al., 2016*). Further, various fungal groups also exhibit the presence of Dicer and plant-like DSRM proteins and lack animal-like accessory proteins, such as Drosha and Pasha (*Dang et al., 2011*; *Dias et al., 2017*). Contrastingly, DCL3 of *Chlamydomonas* exhibits some structural features that are reminiscent of metazoan Drosha (*Valli et al., 2016*). These observations suggest that the common ancestor of all these groups might have harbored only a single Dicer/Drosha-like RNase III enzyme assisted by a DSRM protein resembling the ones found in plants (HYL1-like).

We unexpectedly found that in contrast to plant HYL1 proteins, which interacts with both pri- and pre-miRNA, Hyl1La in *Nematostella* interacts only with pre-miRNA and not with pri-miRNA (*Figure 5D*). Our sequencing data strengthens the specificity of the pre-miRNA binding by showing no enrichment for additional longer RNA molecules (*Figure 5—figure supplement 4A*). Further supporting the specific binding of Hyl1La to pre-miRNAs is the in vitro binding assay to a synthetic pre-miRNA compared to the shuffled sequence with a different secondary structure (*Figure 6D*). A plausible explanation to this finding might be that *Nematostella* already possesses miRNA biogenesis machinery like the Drosha-Pasha microprocessor (*Moran et al., 2013*) that is known to interact only with pri-miRNAs and crop them into the pre-miRNAs (*Kim et al., 2009*). Another surprising finding in this study is that we were able to knock down the Hyl1La by using the MO only and not by shRNA microinjection, despite the processing of the shRNA (*Figure 4D–E*). A possible explanation for this contrasting result between MO and shRNA probably lies between the different mode of action of these two molecules. In contrast to MOs that do not use the cellular machinery, shRNA requires the miRNA/RNAi machinery for their production as well as in target recognition and inhibition. Thus, our combined results suggest that Hyl1La might have an additional effect on biogenesis steps that are downstream to the cleavage by Dicer such as loading of small RNAs into AGO, the protein at the heart of the RISC (*Hutvagner and Simard, 2008*). Under such a condition the shRNA-derived small RNA would be unable to load onto RISC and hence could not cleave the Hyl1La, rendering its expression unaffected. Further, in such a scenario after injection with the shRNAs, the system might reach a balance point that is very close to the normal Hyl1La levels. Alternatively, it is possible that the three shRNAs are ineffective due to lack of accessibility of the three distinct target sites on the Hyl1La transcript to the RISC loaded with the shRNA-derived small RNAs for reasons such as secondary structure or binding by other proteins that restrict the RISC accessibility. However, we find this explanation less likely because the three shRNAs target distinct parts of this relatively long transcript.

During small RNA analysis, we expectedly observed a strong reduction in miRNA expression but in parallel a mild reduction in siRNA expression was also observed in Hyl1La-depleted animals (*Figure 3* and *Figure 3—figure supplement 2A-B*). This might be due to involvement of Hyl1La in siRNA biogenesis pathway in *Nematostella*. Similar observation was also reported in *Arabidopsis* where different members of the DRB protein family, which includes HYL1, are known to be involved in biogenesis of different siRNAs and hinder their expression (*Curtin et al., 2008*; *Raja et al., 2014*). For example, a HYL1 (DRB1) mutant of *Arabidopsis* exhibits severe defects in miRNA expression but also has a partial effect on tasiRNA (a plant-specific category of siRNAs) expression (*Tagami et al., 2009*). Further, DRB2 and DRB4 were shown to inhibit small RNA expression produced by RNA polymerase IV (*Pélissier et al., 2011*). In addition to miRNAs and siRNAs, an opposite trend in piRNA expression was also observed as most of the piRNAs were upregulated in Hyl1La-depleted animals (*Figure 3A* and *Figure 3—figure supplement 2C-D*). We propose this might be due to the fact that depletion of Hyl1La-dependent miRNAs provided a void in the sequencing libraries that in turn made more reads available for piRNAs.

Finally, here we report that Hyl1La plays an important role in miRNA biogenesis in *Nematostella*, a representative of Cnidaria which is the sister group of Bilateria. However, the functional importance of Hyl1La in *Nematostella* identified here raises another interesting evolutionary question of what led to the replacement of Hyl1La by other DSRM proteins like TRBP, Loqs, or PACT in bilaterian animals during evolution (*Figure 1A*). Interestingly, both Loqs in flies and TRBP in mammals enable processing of some miRNA precursors into different mature miRNAs and by this significantly increase their variability and targeted sequences (*Fukunaga et al., 2012*; *Lee and Doudna, 2012*). Such variability is

currently unknown in plants or cnidarians. It is intriguing to consider the possibility that this ability of the bilaterian proteins to increase small RNA variability was advantageous over Hyl1-like proteins and led to the loss of the latter in bilaterian lineages.

## Materials and methods

**Key resources table**

| Reagent type (species) or resource | Designation | Source or reference | Identifiers | Additional information |
|---|---|---|---|---|
| Gene (*Nematostella vectensis*) | Hyl1La | GenBank | KF192067 | |
| Strain, strain background (*Nematostella vectensis*) | Lab strain, Rhode River, MD | Lab strain | | Sea anemone species |
| Strain, strain background (*Escherichia coli*) | NEB 5-alpha Competent *E. coli* (High Efficiency) (DH5α) | New England Biolabs | C2987I | Chemically competent cells |
| Antibody | Monoclonal mouse anti-FLAG M2 antibody (Mouse monoclonal) | Sigma-Aldrich | F1804-50UG | IP (5 µg per test) WB (1:500) |
| Antibody | Peroxidase-AffiniPure Goat Anti-Mouse IgG (Goat polyclonal) | Jackson ImmunoResearch | 115-035-146 | WB (1:10,000) |
| Recombinant DNA reagent | pER242 (plasmid) | *Admoni et al., 2020* | | Used as backbone |
| Sequence-based reagent | PCR Primers | Integrated DNA Technologies | In this paper | See Materials and methods and *Supplementary file 7* |
| Sequence-based reagent | DNA template of shRNA | Integrated DNA Technologies | In this paper | See Materials and methods |
| Sequence-based reagent | Morpholino | Gene Tools | In this paper | See Materials and methods |
| Chemical compound, drug | Trizol | Thermo (Ambion) | 15596026 | |
| Chemical compound, drug | Tryptone | Merck Millipore | 61930505001730 | For bacterial media |
| Chemical compound, drug | Yeast extract purified for Microbiology | Merck Millipore | 61931105001730 | For bacterial media |
| Chemical compound, drug | Agar purified for Microbiology | Merck Millipore | 61939005001730 | For bacterial media |
| Chemical compound, drug | Ampicillin | ROTH | K029.1 | For bacterial media |
| Chemical compound, drug | Dextran Alexa Fluor 488 | Thermo (Molecular Probes) | D22910 | |
| Chemical compound, drug | Red sea salt | Red sea | | For *Nematostella vectensis* growth |
| Chemical compound, drug | L-Cysteine | Merck Millipore | 1028380100 | |
| Chemical compound, drug | Tween20 | Sigma-Aldrich | P9416-100ML | |

*Continued on next page*

*Continued*

| Reagent type (species) or resource | Designation | Source or reference | Identifiers | Additional information |
|---|---|---|---|---|
| Chemical compound, drug | NP40 | Sigma-Aldrich | NP40S-100ML | |
| Chemical compound, drug | Skim milk | BD | 232100 | |
| Chemical compound, drug | Bovine serum albumin (fraction V) | MP | 160069 | |
| Chemical compound, drug | Tris-glycine-SDS buffer | Bio-Rad | 1610772 | For SDS-PAGE |
| Chemical compound, drug | Total mouse IgG | Sigma-Aldrich | I5381-1MG | IP (5 µg per test) |
| Commercial Assay, kit | SuperScript III Reverse Transcriptase | Thermo (Invitrogen) | 18080044 | |
| Commercial Assay, kit | iScript cDNA Synthesis Kit | Bio-Rad | 1708891 | |
| Commercial Assay, kit | Fast SYBR Green Master Mix | Thermo (ABI) | AB-4385612 | |
| Commercial Assay, kit | Q5 High-Fidelity DNA Polymerase | New England Biolabs | M0493S | |
| Commercial Assay, kit | AmpliScribe T7-Flash Transcription Kit | Lucigen | ASF3507 | |
| Commercial Assay, kit | Quick-RNA Miniprep | Zymo Research | R1054 | |
| Commercial Assay, kit | NucleoSpin Gel and PCR Clean-up | Macherey-Nagel | MAN-740609.50 | |
| Commercial Assay, kit | NEBuilder HiFi DNA Assembly Master Mix | New England Biolabs | E2621S | |
| Commercial Assay, kit | CloneJet cloning kit | Thermo (Fermentas) | K1231 | |
| Commercial Assay, kit | HiSpeed Plasmid Midi Kit | Qiagen | 12643 | |
| Commercial Assay, kit | PureLink Quick Plasmid Miniprep Kit | Thermo (Invitrogen) | K210010 | |
| Commercial Assay, kit | NextSeq 500/550v2 Kits (75 cycles) | Illumina | FC-404–2005 | |
| Commercial Assay, kit | NEBNext Multiplex Small RNA Library Prep Set for Illumina (1-12) – 24 rxns | New England Biolabs | NEB-E7300S | |
| Commercial Assay, kit | Pierce RNA 3' End Biotinylation Kit | Thermo Fisher Scientific | 20160 | |
| Commercial Assay, kit | Ovation SoLo RNA-seq systems kit | Tecan Genomics | 0500–32 | |
| Commercial Assay, kit | SureBeads Protein G Magnetic Beads | Bio-Rad | 1614023 | For IP |
| Commercial Assay, kit | Streptavidin Magnetic Beads | New England Biolabs | S1420S | For pull-down |
| Commercial Assay, kit | RNase Inhibitor, Murine | New England Biolabs | M0314L | |

*Continued*

| Reagent type (species) or resource | Designation | Source or reference | Identifiers | Additional information |
|---|---|---|---|---|
| Commercial Assay, kit | cOmplete ULTRA Tablets, Mini, EASYpack Protease Inhibitor Cocktail | Roche | 05892970001 | |
| Commercial Assay, kit | Protease Inhibitor Cocktail Set III, EDTA-Free | Merck-Millipore | 539134–1ML | |
| Commercial Assay, kit | 4–15% Mini-PROTEAN TGX Precast Protein Gels | Bio-Rad | 4561083 | For Western blot |
| Commercial Assay, kit | Trans-Blot Turbo Mini 0.2 µm PVDF Transfer Packs | Bio-Rad | 1704156 | For Western blot |
| Software, algorithm | miRDeep2 | doi:10.1093/nar/gkr688 (2012) | | Small RNA analysis |
| Software, algorithm | The UEA small RNA Workbench | doi:10.1093/bioinformatics/bts311 | | Small RNA analysis |
| Software, algorithm | Trimmomatic (v3.4) | doi:10.1093/bioinformatics/btu170. Epub 2014 Apr 1 | | Total RNA analysis |
| Software, algorithm | STAR (v2.7.9) | doi:10.1093/bioinformatics/bts635 | | Total RNA analysis |
| Software, algorithm | RSEM | doi:10.1186/1471-2105-12-323 | | Total RNA analysis |
| Software, algorithm | shRNA design | doi:10.1016/j.ydbio.2019.01.005 | https://www.invivogen.com/sirnawizard/index | |
| Software, algorithm | Protein Domain Search | https://pfam.xfam.org/search#tabview=t | | |
| Software, algorithm | Homologs search | https://blast.ncbi.nlm.nih.gov/Blast.cgi/Proteins | | |

## Animal culture and microinjection

*Nematostella* were grown in 16‰ artificial salt water (ASW) at 18°C in a dark culture room. The growing animals were fed with freshly hatched *Artemia salina* nauplii three times a week. Induction of spawning was performed as previously described (*Genikhovich and Technau, 2009*): the mature male and female animals were induced to produce eggs and sperm by placing them in an incubator for 8 hr under constant white light and heat (25°C). After induction, the tanks were further kept at 18°C (in the culture room) for 2 hr to allow the release of egg packages and sperm. Further, the egg packages were fertilized for 30 min by placing the packages inside the male tanks. The quality of egg packages was checked under the stereomicroscope and egg packages of round shape and homogenous size were processed further for dejellying using 3% of L-cysteine in 16‰ ASW pH 7.2 (titrated with 10 M NaOH). The selected egg packages were kept in the cysteine solution for 15 min while rotated by hand. The eggs were washed using 16‰ ASW in tissue culture plates. These clean eggs (zygotes) were further used for microinjection. For microinjection 1 mM stock solutions of both MO and shRNA were prepared by dissolving them into nuclease-free water. The toxicity of MOs as well as shRNA was optimized by injecting different concentrations into the animals along with the control-injected animals. Concentrations resulting in toxicity of less than 30% of the animals (estimated morphologically in the 2 days following the injection) were considered suitable for injection. All MOs used in this study were designed and synthesized by Gene Tools, LLC (Philomath, OR, USA). The same control MO was used in all experiments.

    Hyl1La TB MO (Translation Blocking) GGCCGCCATTTCTTAGAGAAGTTCA
    Hyl1La SI MO1 (Splicing inhibition) AGAAACAGACTTGTACCTTTTTGTA
    Hyl1La SI MO2 (Splicing inhibition) CTTGTTGTAGTCTAAGCCTTACCAT
    Dicer1 TB MO (Translation Blocking) ATTCCTCTTCGTCACTTGACATCTT
    Control MO (Standard control MO) CCTCTTACCTCAGTTACAATTTATA

We found that the optimum concentrations were 300, 900, and 450 µM for the Hyl1La TB, Hyl1La SI MO1, and Hyl1La SI MO2, respectively. Dicer1 TB MO was injected at the concentration of 805 µM. For all the three shRNAs, 600 ng/µl concentration was found to be suitable. Similar concentration of control MO was used for microinjection in parallel with Hyl1La MOs. In every shift, we injected 600 zygotes (300 control MO and 300 Hyl1La MO) by mixing the injected material with dextran Alexa Fluor 488 (Thermo Fisher Scientific, Waltham, MA, USA) which was used as a fluorescent tracer while injection was carried under magnification by a TS-100F fluorescent microscope (Nikon). The injected zygotes were kept at 22°C for further growth. The morphology of the animals was observed for up to 9 days after which the number of settled and unsettled animals were counted and documented under SMZ-18 fluorescent stereomicroscope (Nikon). Animals injected with Dicer1 MO were observed and documented after 10 days. For RNA extraction microinjected zygotes were flash-frozen in liquid nitrogen after 3 days of growth and stored at –80°C until RNA extraction. All the MO and shRNA injection experiments were performed in three independent biological replicates with three distinct animals' batches. Dicer1 MO injection was performed in one replicate to repeat an experiment with published results (*Modepalli et al., 2018*).

## Small RNA sequencing and analysis

The RNA was isolated using Trizol (Thermo Fisher Scientific) from 3-day-old animals. Small RNA sequencing was performed for only Hyl1La SI MO1 and control MO-injected animals. The small RNA library was prepared using NEBNext Multiplex Small RNA Library Prep Illumina kit (New England Biolabs, Ipswich, MA, USA) with some modifications (*Plotnikova et al., 2019*). In brief, small RNAs were isolated (18–30 nt) from 1 µg of total RNA using 15% urea-PAGE (Bio-Rad, Hercules, CA, USA) followed by overnight precipitation using 0.3% NaCl. The size-selected small RNAs were further precipitated using ice-cold ethanol (2.5 × volume) and 1 µl of GlycoBlue (Thermo Fisher Scientific) by centrifugation at 21130 × *g*. The pellet was dissolved in 7.5 µl nuclease-free water and used further for adapter ligation. The ligated products were subjected to 14 cycles of PCR amplification using adapter specific primers. The PCR product was run on 2% agarose gel followed by staining with GelRed (Biotium, Fremont, CA, USA). The band size between 137 and 149 nt was selected and purified using Gel Extraction Kit (Macherey-Nagel, Düren, Germany). The quality of the purified product (sRNA-seq libraries) was checked by using TapeStation system (Agilent, Santa Clara, CA, USA). The libraries having a dominant peak at the size range of 137–149 nt were sequenced by NextSeq500 (Illumina) in single-end mode.

The small RNA data was analyzed using miRProf (*Stocks et al., 2012*) with the following parameters: two mismatches allowed, minimum abundance of 1 for miRNA and 5 for piRNA and siRNA, allowed overhang and not grouping mature and star strands. For miRNA, piRNA, and siRNA analysis, we mapped the small RNA sequences after adapter removal on the relevant previously reported sets of *Nematostella* small RNAs (*Fridrich et al., 2020*; *Modepalli et al., 2018*; *Praher et al., 2017*). The small RNA expression was normalized in TPM by using only the transcripts that mapped on the reference genome. For read length distribution and scattered plot we used the average of expression obtained from the three biological replicates. For mapping onto the genome and miRNA precursor for identification of aberrant processing of miRNA, miRDeep2 (*Friedländer et al., 2012*) was used.

Meta-analysis was performed to compare the pattern of miRNA expression following the knockdown of miRNA biogenesis components. Raw reads from small RNA-seq experiments involving the knockdown of various miRNA biogenesis components (Hyl1La, Dicer1, HEN1, AGO1, and AGO2) were downloaded from the NCBI Sequence Read Archive (*Fridrich et al., 2020*; *Modepalli et al., 2018*). Raw reads were trimmed using Cutadapt (v3.4) (*Martin, 2011*) and sequences shorter than 18 nt were discarded. Filtered reads were mapped to the *Nematostella* genome using Bowtie (v1.3.1) (*Langmead et al., 2009*) and miRNA loci quantified using featureCounts (v2.0.0)(*Liao et al., 2014*) from previously characterized coordinates. Due to differences in developmental stages and library preparation, reads were normalized between morphants and their respective control samples. PCA was then performed using these normalized values using ggplot2 in R (v4.1.0).

## Total RNA sequencing and analysis

To generate total RNA libraries from immunoprecipitated samples, RNA was extracted with 1 ml Trizol (Thermo Fisher Scientific) and purified following the manufacturer's protocol. To increase the yield,

we added 1 µl of RNA-grade glycogen (Roche, Basel, Switzerland) into the isopropanol during the precipitation step and conducted two rounds of 75% ice-cold ethanol washes. RNA pellets were dissolved with 8 µl low-EDTA TE buffer (Thermo Fisher Scientific) and diluted with 0.1% Tween-20 (Sigma-Aldrich, Saint Louis, MO, USA). The samples were prepared for sequencing using Ovation SoLo RNA-seq systems kit (Tecan Genomics, Redwood City, USA) according to the manufacturer's protocol with the exclusion of the AnyDeplete stage. The samples were amplified for 13, 14, and 16 cycles using uniquely barcoded adaptors provided in the kit. The quality and size distribution of the produced libraries was checked using TapeStation system (Agilent). The pooled libraries were run on NextSeq500 (Illumina) in the concentration of 1.1 pM.

Raw reads were processed by Trimmomatic (v3.4), using the following parameters (HEADCROP:9 LEADING:3 TRAILING:3 SLIDINGWINDOW:4:20 MINLEN:36) (*Bolger et al., 2014*). High-quality reads retained were mapped to the *Nematostella* genome (JGI.Nemve1) using STAR (v2.7.9) (*Dobin et al., 2013*) and de-duplexed using NuDup (https://github.com/tecangenomics/nudup; software by Tecan Genomics, *Bruns, 2022*). Transcript abundances were estimated to previously characterized coding (*Schwaiger et al., 2014*) and non-coding RNA (*Fridrich et al., 2020*; *Modepalli et al., 2018*) using RSEM (v1.3) (*Li and Dewey, 2011*) and cross-sample normalized using TMM (*Robinson and Oshlack, 2010*).

## Synthesis of shRNA

Potential shRNA precursors for Hyl1La gene were predicted using the shRNA prediction tools (https://www.invivogen.com/sirnawizard/index.php) (*Karabulut et al., 2019*). Three precursors from three different regions were further chosen, all having GC content of more than 35%. Further, we also added to the sequence a T7 promotor and three different mismatches at nucleotide positions 10, 13, and 16 to create bulges in the precursors (*Figure 4A–C*). All these modified precursors were reverse complemented and synthesized at the DNA level by Integrated DNA Technologies, Inc (IDT, Coralville, IA, USA). The DNA templates and reverse primer were mixed (1:1) and denatured at 98 °C for 5 min and cooled to 24 °C. Further, this mixture was mixed with the components of AmpliScribe T7-Flash Transcription Kit (Lucigen, Middleton, WI, USA) and incubated for 8 hr at room temperature. The in vitro transcribed product was further purified using Quick-RNA Miniprep (Zymo Research, Irvine, CA, USA). The quality and size of the precursor was checked on agarose gel and its concentration was measured using Qubit RNA BR (Broad Range) Assay Kit with the Qubit Fluorometer (Thermo Fisher Scientific). The concentration ranged from 1500 to 2000 ng/µl.

## Reverse transcription-quantitative PCR

For the quantification of Hyl1La transcripts from shRNA-injected animals and for checking the splicing inhibition (Hyl1La SI MO-injected animals), cDNA was prepared from 500 ng of total RNA using the iScript cDNA Synthesis Kit (Bio-Rad). For the quantification of miRNAs and shRNA, we designed the stem-loop primers for five different miRNAs and shRNA (*Chen et al., 2005*). For cDNA preparation, 100 ng of total RNA was reverse transcribed using the SuperScript III Reverse Transcriptase (Thermo Fisher Scientific). The specificity of the miRNA primers was determined by using end point PCR (*Varkonyi-Gasic et al., 2007*). For this, we used 2 µl of cDNA as template, miRNAs-specific forward primer and stem-loop-specific reverse primer and run the PCR at 94°C for 2 min, followed by 35 cycles of 94°C for 15 s and 60°C for 1 min. For analyzing differential expression, we ran qRT-PCR with 5sRNA as an internal control. For amplification of pre-miR-2029, pre-miR-2030, and pre-miR-2035–1, we used 1.5 µl of cDNA. For all the real-time experiments, we used Fast SYBR Green Master Mix (Thermo Fisher Scientific) and samples were run on StepOnePlus Real-Time PCR System (Thermo Fisher Scientific). All the real-time experiments were performed in three independent biological replicates and two technical replicates and data was analyzed using $2^{-\Delta\Delta Ct}$ method (*Livak and Schmittgen, 2001*). All the primers are listed in *Supplementary file 7*.

## Cloning and sequencing of Hyl1La SI MO-injected animals

To validate the effect of splicing MO, we designed the primers pairs spanning the introns lying on the boundary of exons. PCR of the Hyl1La was done using Q5 High-Fidelity DNA Polymerase (New England Biolabs). The PCR products were run on the gel and the expected-sized PCR product was purified with a kit. Then the purified PCR products were ligated into the pJet2.1 vector (Thermo Fisher

Scientific) and transformed into the *Escherichia coli* DH5α strain (NEB5α, New England Biolabs). The plasmids were purified by a PureLink miniprep kit (Thermo Fisher Scientific) and outsourced for Sanger sequencing (HyLabs, Israel).

## Plasmid generation

Two gBlock synthetic DNA fragments (IDT) at the lengths of 1.6 and 1.7 kb corresponding to Hyl1La fragments with a 3 × FLAG tag and 20 bp overlaps were ordered and used for generating the expression cassette. These fragments were PCR-amplified by Q5 Hot Start High-Fidelity DNA Polymerase (New England Biolabs), visualized on 1% agarose gel and purified by NucleoSpin Gel and PCR Clean-up (Macherey-Nagel). Gibson assembly was performed with the NEBuilder HiFi DNA Assembly Master Mix (New England Biolabs) following the manufacturer's protocol. The resulting product was further subcloned by restriction digestion with *AscI* and *SalI* into a pER242 vector having a TBP promoter previously proved to drive ubiquitous expression in *Nematostella* (*Admoni et al., 2020*), memOrange2, and SV40 polyadenylation signal (*Figure 5A*). The transformation was performed in *E. coli* DH5α strain (New England Biolabs). The plasmid was purified by HiSpeed Plasmid Midi Kit (Qiagen, Hilden, Germany) and sequenced by the Sanger method (HyLabs, Israel); 100 ng/µl of purified plasmid was injected into the fertilized *Nematostella* embryo and visualized after 2 days under an SMZ18 stereomicroscope equipped with a DS-Qi2 camera (Nikon, Tokyo, Japan).

## Hyl1La IP

One-hundred µl of protein G SureBeads magnetic beads (Bio-Rad) were washed five times with 1 × PBS (phosphate-buffered saline). Five µg of monoclonal mouse anti-FLAG M2 antibody (Sigma-Aldrich) or total mouse IgG (Sigma-Aldrich) were added to the washed beads and incubated overnight at 4°C on a rotating shaker. Three thousand zygotes were injected with the plasmid containing 3 × FLAG-Hyl1La among which ~2000 animals survived after 2 days and were used for protein extraction. Protein was extracted in lysis buffer with the following composition: 25 mM Tris-HCl (pH 7.4), 150 mM KCl, 25 mM EDTA, 0.5% NP-40, 1 mM DTT, Protease inhibitor cOmplete ULTRA tablets (Roche, Germany) and Protease Inhibitor Cocktail Set III, EDTA-Free (Merck Millipore, Burlington, MA, USA). Murine RNase inhibitor (New England Biolabs) was used in RNA processing buffer. The RNase and protease inhibitors were added fresh just before use. For protein extraction, the frozen animals were mechanically homogenized in 1 ml lysis buffer and incubated for rotation at 4°C. After 2 hr the samples were centrifuged at 16000 × *g* for 15 min at 4°C and supernatant was collected. Then, 100 µl of protein G magnetic beads were washed thrice with 1 ml 1 × PBS, once with lysis buffer and then mixed with the protein lysate. The tube volume was maintained to 1.2 ml using the lysis buffer containing RNase inhibitor and incubated at 4°C on a rotating shaker (Intelli-Mixer ELMI, Newbury Park, CA, USA), for 1 hr. After 1 hr, the pre-cleared lysate was collected and added to the antibody-bound beads that were preincubated with the antibody overnight. These samples were incubated for 2 hr in rotation at 4°C. After incubation, the beads were collected by using a magnetic stand and washed six times with lysis buffer containing RNase inhibitor and one time with PBS with RNAse inhibitor. For Western blot 40 µl SDS sample buffer (New England Biolabs) were added to the beads and heated at 100°C for 8 min and placed on ice for 1 min. The samples were then centrifuged for 1 min at 16,000 × *g* at 4°C, and the supernatant was collected.

For RNA extraction, the beads were mixed with 1 ml Trizol (Thermo Fisher Scientific) and purified following the manufacturer's protocol. We added 1 µl of RNA-grade glycogen (Roche, Switzerland) into the isopropanol during the precipitation step. The isolated RNA was treated with Turbo DNAse (Thermo Fisher Scientific) for 30 min at 37°C, purified with RNA Clean & Concentrator-5 kit (Zymo Research), eluted in 8 µl and used for cDNA preparation.

## Primer designing for pre- and pri-miRNA quantification

The pre-miRNA primer pairs were designed from stable stem region of precursors (*Figure 5—figure supplement 2*) as described previously (*Schmittgen et al., 2008*). The pre-miRNA sequence was obtained from our recently published data (*Fridrich et al., 2020*). The primer pairs for pri-miRNA were designed so they will anneal at least 10 nucleotides away from the pre-miRNA primers (*Figure 5—figure supplement 3*). These probable pri-miRNA sequences flanking the pre-miRNA were obtained from the *Nematostella* genome browser (https://simrbase.stowers.org/).

## Biotin-labeled synthetic pre-miRNA generation

Synthetic pre-miRNA for in vitro binding assay was designed based on miR-2022 backbone, with the mature and star stands changed to target an mCherry synthetic gene, which is not found in the *Nematostella* genome. To generate a control sequence, the original sequence was shuffled and non-hairpin secondary structure was validated with RNAfold web server (*Lorenz et al., 2011*). The reverse complement DNA templates were ordered from IDT (USA) and transcribed using the AmpliScribe T7-Flash Transcription kit protocol (Lucigen). The DNase-treated products were cleaned with Quick-RNA MiniPrep Kit (Zymo Research), then validated on agarose gel and concentration was measured with Epoch Microplate Spectrophotometer (BioTek Instruments, Inc, USA). Next, 790 ng of the products were biotinylated with Pierce RNA 3' End Biotinylation Kit (Thermo Fisher Scientific). After ligation of 15 hr, products were cleaned according to the manufacturer's protocol and pellets were dissolved with 12.5 µl nuclease-free water. A second cleaning step was conducted with Quick-RNA MiniPrep Kit (Zymo Research). Products were eluted with 16 µl nuclease-free water and concentration was measured with NanoDrop (Thermo Fisher Scientific).

## In vitro binding assay

In vitro binding assay was performed according to a previously described method (*Lewandowska et al., 2021*). In brief, 4000 zygotes were injected with the plasmid containing 3 × FLAG-Hyl1La, from which ~3000 survived after 2 days. Embryos were flash-frozen with liquid nitrogen and stored in –80°C. Next, animals were mechanically homogenized in the following lysis buffer: 50 mM Tris-HCl (pH 7.4), 150 mM KCl, 0.5% NP-40, 10% glycerol, Protease inhibitor cOmplete ULTRA tablets (Roche, Switzerland), Protease Inhibitor Cocktail Set III, EDTA-Free (Merck Millipore), and Murine RNase inhibitor (New England Biolabs). Protease and RNase inhibitors were added fresh just before use. Lysed animals were incubated and supernatants were collected as described before. Next, the lysate was pre-cleared as follows: 100 µl of streptavidin magnetic beads (New England Biolabs) were washed in 1 ml of 1 × PBS for three times and the FLAG-tagged Hyl1La lysate was added to the washed beads. Lysis buffer was added to make up 1.3 ml and samples were incubated at 4°C rotation for 1 hr. After the incubation, the pre-cleared lysates were collected and mixed with the biotin-labeled pre-miRNA or biotin-labeled shuffled pre-miRNA as negative control in the final concentration of 8.23 ng/ml and ATP (New England Biolabs) in the final concentration of 0.5 mM. Samples were incubated for 1 hr in rotation at room temperature. Simultaneously, 100 µl of fresh streptavidin magnetic beads were washed as described before and blocked with 3-day-old wild-type lysate for 1 hr at 4°C. Biotin-labeled precursors containing lysates were added to the blocked beads and incubated for 2 hr in rotation at 4°C for biotin-labeled pre-miRNA pull-down. Sixty µl was taken from each lysate before addition to the beads as input sample. After the incubation, the lysates were discarded and the beads were washed three times with 500 µl of the following wash buffer: 50 mM Tris-HCl (pH 7.4), Protease inhibitor cOmplete ULTRA tablets, Protease Inhibitor Cocktail Set III, EDTA-Free and Murine RNase inhibitor. Subsequently, 40 µl of filtered double-distilled water and 20 or 30 µl of Blue Protein Loading Dye (New England Biolabs) were added to the beads or the inputs, respectively. The samples were heated and centrifuged as described before and the supernatant was collected for Western blot. Intensities of Western bands were determined by using the ImageStudio software (LI-COR Biosciences, Lincoln, NE, USA). Enrichment of pre-miRNA pull-down compared to shuffled pre-miRNA pull-down was determined by the ratio of the band to the background intensity.

## Quantification and statistical analysis

For phenotypic analysis, we performed Student's t-test between the number of developed and undeveloped animals. To test differences in size distribution of small RNA sequencing reads, Student's t-test (paired two-tailed) was performed between control and Hyl1La SI MO1 samples. For statistical analysis of qRT-PCR data, Student's t-test (one-tailed assuming equal variance) was performed on ΔCt values between different biological replicates. For in vitro binding assay analysis, Student's t-test (paired two-tailed) was performed on the ratio of the bands to background intensity. To check overall significant difference between the miRNA expression levels, Wilcoxon signed-rank test was done. The Student's t-test was conducted in Microsoft Excel while Wilcoxon signed-rank test was done using socscistatistics (https://www.socscistatistics.com/tests/signedranks/default.aspx).

## Acknowledgements

The authors would like to thank Dr Michal Bronstein and Mrs Adi Turjeman of the Centre for Genomic Technologies (The Hebrew University) for their help with sequencing.

## Additional information

### Funding

| Funder | Grant reference number | Author |
|--------|------------------------|--------|
| H2020 European Research Council | 637456 | Yehu Moran |
| H2020 European Research Council | 863809 | Yehu Moran |

The funders had no role in study design, data collection and interpretation, or the decision to submit the work for publication.

### Author contributions

Abhinandan M Tripathi, Conceptualization, Data curation, Formal analysis, Investigation, Methodology, Validation, Visualization, Writing – original draft; Yael Admoni, Data curation, Formal analysis, Investigation, Methodology, Visualization, Writing – original draft; Arie Fridrich, Magda Lewandowska, Formal analysis, Investigation, Validation, Visualization, Writing – review and editing; Joachim M Surm, Data curation, Formal analysis, Investigation, Visualization, Writing – review and editing; Reuven Aharoni, Investigation, Writing – review and editing; Yehu Moran, Conceptualization, Funding acquisition, Project administration, Resources, Supervision, Writing – original draft

### Author ORCIDs

Yehu Moran (iD) http://orcid.org/0000-0001-9928-9294

### Decision letter and Author response

Decision letter https://doi.org/10.7554/eLife.69464.sa1
Author response https://doi.org/10.7554/eLife.69464.sa2

## Additional files

### Supplementary files

- Supplementary file 1. The morpholino (MO) clones that are showing the intron retention.
- Supplementary file 2. MicroRNAs (miRNAs) and their expression.
- Supplementary file 3. MicroRNAs (miRNAs) and their expression for meta-analysis.
- Supplementary file 4. Pre-microRNA (miRNA) and pri-miRNA Ct values.
- Supplementary file 5. RNA-seq.
- Supplementary file 6. In vitro binding assay band intensities.
- Supplementary file 7. Primers used in this study.
- Transparent reporting form
- Source data 1. Source data file of gels and blots with relevant bands labelled.
- Source data 2. Source data of original gels and blots.

### Data availability

RNA sequencing data are available at NCBI-SRA under BioProject ID PRJNA630340.

The following dataset was generated:

| Author(s) | Year | Dataset title | Dataset URL | Database and Identifier |
|---|---|---|---|---|
| Tripathi AM, Admoni Y, Fridrich A, Lewandowska M, Surm JM, Aharoni R, Moran Y | 2022 | Functional characterization of a plant-like HYL1 homolog in the cnidarian *Nematostella vectensis* indicates a conserved involvement in microRNA biogenesis | https://www.ncbi.nlm.nih.gov/bioproject/PRJNA630340 | NCBI BioProject, PRJNA630340 |

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
