## [Editor Report]

This paper will be of importance for researchers in the field of RNA biology and evolutionary biology. It provides a new perspective on the origins of the miRNA pathways, and proposes a common origin of plant and animal miRNA pathways. The main conclusions of the paper are well supported.

---

## [Decision Letter]

**Decision letter after peer review:**

Thank you for submitting your article "Functional characterization of a "plant-like" HYL1 homolog in the cnidarian *Nematostella vectensis* indicates a conserved involvement in microRNA biogenesis" for consideration by *eLife*. Your article has been reviewed by 3 peer reviewers, including Rene F Ketting as the Reviewing Editor and Reviewer #1, and the evaluation has been overseen by a Reviewing Editor and Detlef Weigel as the Senior Editor.

Essential revisions:

1) As pre-discussed, a strengthening of the claim that Hyl1La specifically binds miRNA precursors is needed. If a CLIP experiment is not feasible, a qPCR approach testing more miRNA precursors, but more importantly, non-related transcripts would also be acceptable. We would see testing of 10-12 mRNAs, 10-12 structured RNAs (for instance tRNAs) and a similar number of additional miRNA precursors as an appropriate alternative.

2) The small RNA analysis needs to be extended to also include piRNAs and siRNAs.

*Reviewer #1:*

This work addresses the function of a Hyl1 homolog of Nematostella in miRNA production. Hyl1 is a protein that was identified in plants, where it stimulates the processing of miRNAs. In animals, Hyl1 is not found, and other RNA binding proteins are in place to help processing of miRNAs by Drosha and Dicer. Linking Hyl1 in Nematostella to miRNA production would support the idea that in the last common ancestor of plants and animals a miRNA-like processing was in place. This contrasts the dominating view that plant and animal miRNA pathways have convergently evolved and would justify a revised view on how plant and animal miRNA pathways have evolved.

To test this, Hyl1La (the Nematostella Hyl1) was knocked down with morpholino oligos (MOs), and the resulting animals were probed for developmental defects and miRNA production. In both defects were observed. Animal development was delayed, and miRNA production dropped. Using RNA-IP-qPCR experiments the authors also address whether Hyl1La binds primary miRNAs or pre-miRNAs, two different maturation steps in the animal miRNA processing pathway. While pri-miRNAs were hardly detected in the IPs, pre-miRNAs were enriched, indicating that Hyl1La binds the intermediate miRNA processing stage, in between Drosha and Dicer cleavages.

The authors also attempt knock-down of Hyl1La by shRNAs but were not successful. This lack of success is hypothesized to relate to a potential role of Hyl1La in shRNA function.

The paper is rather short, but it makes the point that Hyl1La is needed for miRNA production, and as such brings an important piece of data to the discussion on miRNA pathway evolution in plants and animals. Nematostella is also not a model in which extensive manipulations of gene function are easily done, limiting what the authors can realistically do.

Parts of the paper are, however, not completely clear and in one case not consistent with the data that is shown:

– In Figure 3-supplement 1 two additional MOs are tested for knock-down. In the text a translation blocking MO is mentioned (TB) but this does not appear in this figure. It is also not clear if the same control MO is used for this figure compared to Figure 3. And finally, the text states that in this experiment mir-2026-5p was consistent down regulated, but MO1 does not show a significant effect in the figure.

– The authors do not discuss the miRNAs that are not affected.

– The authors mention that Hyl1 in plants is required for accuracy (page 4), but later (page 5) they write "The analysis (of Hyl1La, red) did not reveal any aberrant processing. These results further suggested that like its homolog in plants, Hyl1La in Nematostella…//…is not involved in size selection." These statements appear to be contradictory, assuming that with 'accuracy' miRNA processing sites are implied. This issue also surfaces in the discussion.

Finally, the developmental defects are not linked to miRNA defects directly, but they do phenocopy knock-down phenotypes of other miRNA pathway components, such as Dicer.

In order to see the similarity of the MO knockdown of Hyl1La it would be good to do a knockdown of Dicer in parallel and compare the phenotypes. Also, the phenotypes shown are at a very zoomed-out level, and some more detail, or more informative images would be helpful.

Please include scale bars for all the imaging.

More importantly, the authors propose a role for binding pre-miRNA, and it would be good to substantiate that more.

One thing that could be interesting, and should be relatively straight forward, is to look at the subcellular localization of Hyl1La. One would expect a cytoplasmic localization if it acts with Dicer.

Maybe cytoplasmic lysate can process pre-miRNAs, and this may be impaired in extracts from knock-down animals. This depends on whether this is feasible in terms of amount of material; I cannot judge that, but I can see that may be difficult.

Third, a Northern blot may help to detect accumulating pre-miRNAs in the knock-down animals (with Dicer knockdown as a control).

In the discussion the authors dedicate a paragraph to basically a failed experiment: the shRNA knockdown. The model behind that discussion is interesting, but should be tested in my opinion: are the established shRNA knockdowns (that are cited) affected by MO knockdown of Dicer (positive control) and Hyl1La?

*Reviewer #2:*

The authors aimed to dissect the miRNA-processing capacity of Nv Hyl1La. The strength of the study is based on the observed developmental defects accompanied by molecular data upon Hyl1La KO. Information about the miRNA target genes that are affected by the Hyl1La-KO are missing as well as biochemical properties of Hyl1La. Working with non-traditional model organisms is challenging but provide a powerful resource for identifying new molecular pathways or revise current perceptions of regulatory mechanisms (i.e., miRNA processing). Cnidaria resemble plant-like miRNA-processing propensities, which have been eluted to in previous studies of the group.

Strength:

– Description of organismal defects upon Hyl1La depletion.

– Successful Hyl1La knockdown by using morpholinos but not shRNAs.

– sRNAseq on Hyl1La knockdown (vs. wild-type).

– Mechanistic information about Hyl1La’s miRNA processing capacity showing that Hyl1La processes pre- but not pri-miRNA (using RIP on FLAG-tagged Hyl1La followed by RT-qPCR)

Weaknesses:

– What are the expression levels of Hyl1Lb upon Hyl1La KD? The authors state that Hyl1Lb is cell-type specifically expressed, which suggests that the role of Hyl1Lb is negligible in this context. That can be easily checked by qPCR using gene-specific primers and would confirm that there are no compensatory effects (e.g., an unexpected gene activation of Hyl1Lb upon Hyl1La KD).

– Missing explanation why only a few miRNAs were detectable in Nv by using sRNAseq. Is this a technical (e.g., low input material, missing step to demethylate 3'ends of Nv sRNAs) or biological (e.g., primitive miRNA machinery with only a few genes that would give rise to miRNAs) issue?

– More comprehensive analysis of the sRNAseq data is required. Information is missing regarding the RNA type to which the entire sRNA population maps. That would explain the shift of sRNA population from 22nt to 28-30nt (Figure 3A) and also why some sRNAs remain unaffected by the Hyl1La-KD (Figure 3B – those on the regression line), e.g. these larger sRNAs could be tRNA or rRNA fragments that are processed in a Hyl1La-independent manner.

– Statements about the comparisons regarding plant HYL1 processing needs to be corrected. E.g. Wu et al. 2007 (doi.org/10.1105/tpc.106.048637) stated that AtHYL1 is involved in pre-miRNA processing.

*Reviewer #3:*

– A summary of what the authors were trying to achieve.

In this manuscript the authors attempt to understand the function of a protein, HYL1, in nematostella. This protein is interesting because HYL1 is known to be involved in miRNA biogenesis in plants, but is absent from most animals (and all bilatarians). The authors wanted to know whether it was involved in miRNA biogenesis in nematostella, as this would shed important light on the evolution of miRNA biogenesis. They investigate the phenotypic consequences of gene knockdown and then attempt to link these to changes in the small RNA profile, and on the basis of their findings argue for a role for HYL1 in miRNA biogenesis in nematostella.

– An account of the major strengths and weaknesses of the methods and results.

The authors use knockdown to reduce HYL1 and show that there is an embryonic lethality phenotype. This suggests importance of the protein in development. These experiments are done with several different knockdown methods, thus meaning that off-target effects are unlikely. Nevertheless, these experiments would be cleaner with deletion of the gene, which presumably is not possible currently in this organism.

The embryonic lethality phenotype is clearly in favour of an important role for this protein. However, it precludes the authors' examining potential functions for HYL1 in adult physiology, which would be interesting and would also be helpful in demonstrating that the role of HYL1 in miRNA biogenesis is direct.

The authors use small RNA sequencing to investigate whether small RNAs are perturbed.

They focus on miRNAs and show that there is a reduction in miRNA levels. This is convincing. However, the authors do not (i) link this to changes in gene expression which might be potentially responsible for the developmental phenotype. (ii) make any analysis of other types of small RNAs.

I would argue that both points are important for the authors' case that HYL1 is a miRNA processing factor that was present in the last eukaryotic common ancestor. It may be that HYL1 is a general DCR partner protein in nematostella and has thus evolved its function in miRNA biogenesis convergently. Furthermore, crucially, the strong developmental phenotype means that changes in miRNAs are possibly secondary to a different function, for example general RNA binding (see below) or a function in endogenous siRNA biogenesis separate from miRNA biogenesis.

The authors experiments that demonstrate binding of the HYL1 to pre-miRNA are useful in that they show HYL1 binds to its hypothesized substrate on the basis of extrapolation from Arabidopsis. However, the authors do not rule out the possibility that HYL1 is a generalized RNA binding protein- this could be tested by some kind of CLIP-seq experiment to assess in an unbiased way the RNA that HYL1 binds to.

– An appraisal of whether the authors achieved their aims, and whether the results support their conclusions.

The authors have certainly demonstrated that miRNAs are affected when HYL1 is removed, but they have not unambiguously determined that HYL1 is directly involved in miRNA processing. Additionally, they have not linked the developmental phenotype to the miRNA processing defects, which means that there is a possibility that the miRNA affects are secondary to another effect of HYL1 on development that alters the miRNA profile profoundly (as many strong developmental perturbations are likely to do).

– A discussion of the likely impact of the work on the field, and the utility of the methods and data to the community.

The demonstration that HYL1 in nematostella is a genuine and specific miRNA processing factor would be really important in redefining how we think of miRNA evolution. It would also be a really striking example of how many mechanisms present in the common ancestor of eukaryotes have been lost in many different lineages. It would therefore be worth this study going a little further to demonstrate that the processing function is direct and specific for miRNAs.

1) Direct demonstration that HYL1 processes miRNAs.

I think that this would be a really important aspect to try to improve in the current manuscript and have two possible suggested experiments:

i) If, instead of knocking down early in development leading to lethality, the authors could introduce knockdown morpholinos later, it might be possible to circumvent the lethality as the developmental problems will be avoided. Then the authors could examine miRNA precursors and mature miRNAs by deep sequencing in organisms over a timecourse of knockdown and potentially without such severe phenotypic effects. They could then be more confident that the influence of HYL1 on miRNA levels is direct.

I appreciate that this may not be technically possible- I am not familiar with the experimental system.

ii) CIPseq against HYL1 to assess in an unbiased way the RNAs that bind- present experiments used qPCR to test presence of miRNA precursors in the IP but this does not rule out the possibility that HYL1 is a general RNA binding protein- there may be other targets that are not directly investigated. Since the authors have already got IP-qPCR to work this should be relatively straightforward to implement. However, the authors should of course not use standard small RNA library preparations from IP RNA because the species will be too long- they should simply use everything and send for a traditional non-polyA total RNAseq. A mock IP without the anti-HYL1 antibody would be important as a control to assess enrichment. This experiment would be a tremendous addition to the paper.

2) A clear link between miRNA dysfunction and developmental phenotype.

This would be a complementary way to try to demonstrate that the effect of HYL1 knockdown on development is through miRNAs. If the authors perform RNAseq over a timecourse of normal and HYL1 knockdown development, they could compare the two to find genes that are differentially expressed. These genes might be miRNA targets, and this could be associated using bioinformatic analysis to test whether the changes are consistent with the alterations in miRNAs observed upon HYL1 knockdown.

3) Another important thing to look at is to examine whether endogenous small RNAs are affected by the knockdown. The authors' model that this is a specific miRNA processing factor would indicate that other classes of small RNAs, including piRNAs and endosiRNAs, will be unaffected or minimally affected. The authors have already generated small RNA sequencing data so this would be straightforward to test.

– Recommendations for improving the writing and presentation.

I'm not sure that the authors should make the bold claim of common evolutionary history of plant and animal miRNA biogenesis in the introduction and abstract. I fully agree that this is an exciting implication, but the paper so far has not demonstrated this- if they were to do some of the experiments above perhaps this would be justified but otherwise it should be placed in the discussion but not elsewhere.

---

## [Author Response]

Essential revisions:1) As pre-discussed, a strengthening of the claim that Hyl1La specifically binds miRNA precursors is needed. If a CLIP experiment is not feasible, a qPCR approach testing more miRNA precursors, but more importantly, non-related transcripts would also be acceptable. We would see testing of 10-12 mRNAs, 10-12 structured RNAs (for instance tRNAs) and a similar number of additional miRNA precursors as an appropriate alternative.

Following the request, we have added qPCR results of three additional miRNA precursors (Figure 5). We understand that this might look like a limited response in light of the reviewers' request for 10-12 miRNA precursors; unfortunately, due to low levels of expression of most miRNAs at the planula life stage in combination with low RNA amounts in immunoprecipitation samples and predicted precursor sequences, we managed to amplify at reasonable levels (i.e., Ct value in the accepted range) only three additional miRNA precursors. miR-2030 and miR-2035-1 show significantly higher expression in Hyl1La immunoprecipitated samples compared to IgG control. In contrast, miR-2029 does not show a significant difference. This can be explained by the expression of miR-2029 in Hyl1La depleted animals that suggests it might not be dependent on Hyl1La for its biogenesis. This notion is further supported by previous results from our lab suggest that miR-2029 responds differently compared to other miRNAs to Dicer1 knockdown in *Nematostella* (Modepalli et al., 2018, PLOS Genet., 14(8): e1007590), which could point to difference in its biogenesis pathway.

In addition to the three miRNA precursors we managed to test, we attempted to amplify miR-2028, miR-2040b, miR-100 and miR-9414 but were unsuccessful.

We also added RNA sequencing information from immunoprecipitated samples that shows no enrichment for rRNA, mRNA ncRNA and repeat elements in the anti-FLAG samples compared to the IgG control samples (Figure 5 —figure supplement 4A). These results further support the specificity of Hyl1La to miRNA precursors. Unfortunately, due to size distribution limitations of the kit at the heart of this method (SoLo by Tecan; selected for its suitability for ultra-low starting material) it could not cover pre-miRNAs (Figure 5 —figure supplement 4B).

In an additional experiment, we show specific binding of Hyl1La to pre-miRNA by in vitro binding assay (Figure 6). We used miR-2022 backbone to generate a synthetic pre-miRNA that mimics with high similarity the structure of an endogenous pre-miRNA. The targeted sequence was altered to not match any *Nematostella* genes but mCherry sequence instead. As negative control we used a shuffled version of the synthetic pre-miRNA that does not generate a hairpin secondary structure. Both pre-miRNAs were biotinylated at their 3' end and incubated with lysate containing the Hyl1La-FLAG tagged protein that was over-expressed in injected animals. Pull-down with streptavidin beads followed by Western blot revealed that Hyl1La binds the pre-miRNA hairpin with significantly higher efficiency than the shuffled sequence (Figure 6 C-D), hence supporting the previous results that Hyl1La binds pre-miRNAs in *Nematostella* in a specific manner.

2) The small RNA analysis needs to be extended to also include piRNAs and siRNAs.

We extended that small RNA analysis to include piRNAs and siRNAs (Figure3 —figure supplement 2). We observed a mild downregulation of siRNAs with 9% of the siRNAs downregulated more than two-fold in HYL1La MO injected animals. piRNAs showed upregulation in Hyl1La depleted animals with 36% of piRNAs upregulated more than two-fold. The effect of Hyl1La depletion on siRNAs could be explained by the involvement of Hyl1La in the siRNA biogenesis pathway in *Nematostella*. Similar effects of double-stranded RNA binding proteins knockdown on siRNAs levels were reported in *Arabidopsi*s (Curtin et al., 2008, FEBS Letters, 582(18): 2753-60; Raja et al., 2014, Journal of Virology, 88(5): 2611-22). Regarding the upregulation of piRNAs, that could be a result of a sequencing void caused by the depletion of miRNAs and siRNAs, which left more space for piRNAs clustering and representation in the libraries.

Reviewer #1:This work addresses the function of a Hyl1 homolog of Nematostella in miRNA production. Hyl1 is a protein that was identified in plants, where it stimulates the processing of miRNAs. In animals, Hyl1 is not found, and other RNA binding proteins are in place to help processing of miRNAs by Drosha and Dicer. Linking Hyl1 in Nematostella to miRNA production would support the idea that in the last common ancestor of plants and animals a miRNA-like processing was in place. This contrasts the dominating view that plant and animal miRNA pathways have convergently evolved and would justify a revised view on how plant and animal miRNA pathways have evolved.To test this, Hyl1La (the Nematostella Hyl1) was knocked down with morpholino oligos (MOs), and the resulting animals were probed for developmental defects and miRNA production. In both defects were observed. Animal development was delayed, and miRNA production dropped. Using RNA-IP-qPCR experiments the authors also address whether Hyl1La binds primary miRNAs or pre-miRNAs, two different maturation steps in the animal miRNA processing pathway. While pri-miRNAs were hardly detected in the IPs, pre-miRNAs were enriched, indicating that Hyl1La binds the intermediate miRNA processing stage, in between Drosha and Dicer cleavages.The authors also attempt knock-down of Hyl1La by shRNAs but were not successful. This lack of success is hypothesized to relate to a potential role of Hyl1La in shRNA function.The paper is rather short, but it makes the point that Hyl1La is needed for miRNA production, and as such brings an important piece of data to the discussion on miRNA pathway evolution in plants and animals. Nematostella is also not a model in which extensive manipulations of gene function are easily done, limiting what the authors can realistically do.Parts of the paper are, however, not completely clear and in one case not consistent with the data that is shown:– In Figure 3-supplement 1 two additional MOs are tested for knock-down. In the text a translation blocking MO is mentioned (TB) but this does not appear in this figure. It is also not clear if the same control MO is used for this figure compared to Figure 3. And finally, the text states that in this experiment mir-2026-5p was consistent down regulated, but MO1 does not show a significant effect in the figure.

The translation blocking (TB) MO knockdown effect on miRNAs is shown in Figure 3 panel D. The two additional MOs tests appear in Figure 3 – supplement 1 as the reviewer mentioned, and the figure legends describe it appropriately. The Control MO we used was the same for all experiments. We added a clarification for this point in the Materials and methods section (Lines 329-330). Regarding miR-2026 we edited the text (Line 150) following the reviewer's comment.

– The authors do not discuss the miRNAs that are not affected.

We expanded the discussion on the miRNAs that are not affected in lines 154-155.

– The authors mention that Hyl1 in plants is required for accuracy (page 4), but later (page 5) they write "The analysis (of Hyl1La, red) did not reveal any aberrant processing. These results further suggested that like its homolog in plants, Hyl1La in Nematostella…//…is not involved in size selection." These statements appear to be contradictory, assuming that with 'accuracy' miRNA processing sites are implied. This issue also surfaces in the discussion.

We thank the reviewer for the comment. We edited the text in lines 133-135 and added a reference (Szarzynska et al., 2009, Nucleic Acids Res. 37(9)) to emphasize that miRNAs in plants mostly show downregulation following Hyl1 knockdown and only a few of them show aberrant processing.

Finally, the developmental defects are not linked to miRNA defects directly, but they do phenocopy knock-down phenotypes of other miRNA pathway components, such as Dicer.

We agree with the reviewer and believe we conveyed this idea in the text (lines 123-125).

In order to see the similarity of the MO knockdown of Hyl1La it would be good to do a knockdown of Dicer in parallel and compare the phenotypes. Also, the phenotypes shown are at a very zoomed-out level, and some more detail, or more informative images would be helpful.

Following the reviewer's request, we performed knockdown of Dicer1 via Morpholino microinjection and documented the phenotype at 10 days post injection. The microscopy pictures were added to Figure 2. The observed phenotype is similar to what is seen in Hyl1La depleted animals, which do not undergo metamorphosis. In addition, the phenotype is similar to previous knockdown experiments of Dicer1 that were performed by our lab members and previously published (Modepalli et al., 2018, PLOS Genet. 14(8): e1007590).

Moreover, we added a principal component analysis (PCA) describing a meta-analysis comparing the knockdown effect of other miRNA biogenesis components on miRNA profiles (Figure 2G). We show that Hyl1La morphants cluster with Dicer1 and HEN1 morphants, suggesting they have a similar effect on a similar subset of miRNAs. In accordance with this result, AGO1 and AGO2 morphants are the most distant from the other components, which is consistent with their function in loading instead of processing miRNAs. The results from this analysis further support the role of Hyl1La in miRNA biogenesis in a similar manner to Dicer1 and HEN1. Sequencing data from the different knockdown experiments were taken from Modepalli et al., 2018, PLOS Genetics 14(8): e1007590 and Fridrich et al., 2020, Nature Communications, 11(1), 1-12. Notably, the clustering occurs despite the fact the libraries in the different studies were generated with different kits and protocols.

Please include scale bars for all the imaging.

Scale bars were added.

More importantly, the authors propose a role for binding pre-miRNA, and it would be good to substantiate that more.One thing that could be interesting, and should be relatively straight forward, is to look at the subcellular localization of Hyl1La. One would expect a cytoplasmic localization if it acts with Dicer.Maybe cytoplasmic lysate can process pre-miRNAs, and this may be impaired in extracts from knock-down animals. This depends on whether this is feasible in terms of amount of material; I cannot judge that, but I can see that may be difficult.

We thank the reviewer for the comment. Please see above our reply to essential revisions section 1. Looking at the subcellular localization is an interesting idea, however, use of this method is currently unavailable in *Nematostella* and we believe that it is beyond the scope of this paper. Testing the processing of pre-miRNAs by cytoplasmatic lysate would indeed be challenging due to limitations in the number of injected animals.

Third, a Northern blot may help to detect accumulating pre-miRNAs in the knock-down animals (with Dicer knockdown as a control).

Limited amount of pre-miRNA in planulae make the suggested experiment very challenging and we believe it is not mandatory to perform in order to support our claims.

In the discussion the authors dedicate a paragraph to basically a failed experiment: the shRNA knockdown. The model behind that discussion is interesting, but should be tested in my opinion: are the established shRNA knockdowns (that are cited) affected by MO knockdown of Dicer (positive control) and Hyl1La?

We thank the reviewer for the suggestion. Unfortunately, such an experiment would be difficult to perform. First, Dicer and Hyl1La are both maternally deposited in *Nematostella*. Second, MO injection might trigger immune response as was shown for several bilaterian species such as fish and frogs and this might increase Argonaute activity.

Reviewer #2:The authors aimed to dissect the miRNA-processing capacity of Nv Hyl1La. The strength of the study is based on the observed developmental defects accompanied by molecular data upon Hyl1La KO. Information about the miRNA target genes that are affected by the Hyl1La-KO are missing as well as biochemical properties of Hyl1La. Working with non-traditional model organisms is challenging but provide a powerful resource for identifying new molecular pathways or revise current perceptions of regulatory mechanisms (i.e., miRNA processing). Cnidaria resemble plant-like miRNA-processing propensities, which have been eluted to in previous studies of the group.Strength:– Description of organismal defects upon Hyl1La depletion.– Successful Hyl1La knockdown by using morpholinos but not shRNAs.– sRNAseq on Hyl1La knockdown (vs. wild-type).– Mechanistic information about Hyl1La’s miRNA processing capacity showing that Hyl1La processes pre- but not pri-miRNA (using RIP on FLAG-tagged Hyl1La followed by RT-qPCR)Weaknesses:– What are the expression levels of Hyl1Lb upon Hyl1La KD? The authors state that Hyl1Lb is cell-type specifically expressed, which suggests that the role of Hyl1Lb is negligible in this context. That can be easily checked by qPCR using gene-specific primers and would confirm that there are no compensatory effects (e.g., an unexpected gene activation of Hyl1Lb upon Hyl1La KD).

Hyl1Lb is expressed in nematocytes and not throughout the planulae like Hyl1La (Moran et al., 2013, Molecular Biology and Evolution, 30(12): 2541-52). Hence, we believe it is not relevant to test for compensatory effect by it. In addition, since we confirmed through sequencing that most miRNA levels are significantly decreased upon Hyl1La knockdown, it is most likely that there is no compensation by Hyl1Lb for the depletion of Hyl1La.

– Missing explanation why only a few miRNAs were detectable in Nv by using sRNAseq. Is this a technical (e.g., low input material, missing step to demethylate 3'ends of Nv sRNAs) or biological (e.g., primitive miRNA machinery with only a few genes that would give rise to miRNAs) issue?

The topic of miRNA expression in the early stages of *Nematostella* life cycle has been thoroughly studied before (Grimson et al., 2008, Nature 455: 1193–1197; Moran et al., 2014, Genome Research, 24(4): 651-663; Modepalli et al., 2018, PLOS Genetics 14(8): e1007590). miRNA expression is difficult to identify without first performing Ago immunoprecipitation (Fridrich et al., 2020, Nature communications, 11(1): 1-12), while the class of piRNAs is very abundant. In addition, the miRNAs that are expressed at this stage are at low levels (Moran et al., 2014, Genome Research, 24(4): 651-663; Modepalli et al., 2018, PLOS Genetics 14(8): e1007590).

– More comprehensive analysis of the sRNAseq data is required. Information is missing regarding the RNA type to which the entire sRNA population maps. That would explain the shift of sRNA population from 22nt to 28-30nt (Figure 3A) and also why some sRNAs remain unaffected by the Hyl1La-KD (Figure 3B – those on the regression line), e.g. these larger sRNAs could be tRNA or rRNA fragments that are processed in a Hyl1La-independent manner.

Following the reviewer's comment, we added analysis of piRNAs (28-30 nt), which is described in section 2 of the essential revisions. piRNAs are highly abundant in *Nematostella* and can be found in somatic cells in early developmental stages (Praher et al., 2017, RNA Biology, 14(12): 1727-1741). Regarding unaffected sRNAs, as mentioned in section 1 of our reply to the essential revisions and in the paper, some miRNAs such as miR-2029 might be Dicer1- and Hyl1La-independent in their biogenesis process.

– Statements about the comparisons regarding plant HYL1 processing needs to be corrected. E.g. Wu et al. 2007 (doi.org/10.1105/tpc.106.048637) stated that AtHYL1 is involved in pre-miRNA processing.

We are not sure we understand what the reviewer means exactly in their comment. We do not imply in the paper that AtHYL1 is not involved in the processing of pre-miRNAs. To the best of our knowledge, HYL1 in plants is involved both in cropping of pri-miRNAs and dicing of pre-miRNAs. We made sure again that this is the message we deliver in our manuscript, and we added the reference mentioned by the reviewer.

Reviewer #3:[…]1) Direct demonstration that HYL1 processes miRNAs.I think that this would be a really important aspect to try to improve in the current manuscript and have two possible suggested experiments:i) If, instead of knocking down early in development leading to lethality, the authors could introduce knockdown morpholinos later, it might be possible to circumvent the lethality as the developmental problems will be avoided. Then the authors could examine miRNA precursors and mature miRNAs by deep sequencing in organisms over a timecourse of knockdown and potentially without such severe phenotypic effects. They could then be more confident that the influence of HYL1 on miRNA levels is direct.I appreciate that this may not be technically possible- I am not familiar with the experimental system.ii) CIPseq against HYL1 to assess in an unbiased way the RNAs that bind- present experiments used qPCR to test presence of miRNA precursors in the IP but this does not rule out the possibility that HYL1 is a general RNA binding protein- there may be other targets that are not directly investigated. Since the authors have already got IP-qPCR to work this should be relatively straightforward to implement. However, the authors should of course not use standard small RNA library preparations from IP RNA because the species will be too long- they should simply use everything and send for a traditional non-polyA total RNAseq. A mock IP without the anti-HYL1 antibody would be important as a control to assess enrichment. This experiment would be a tremendous addition to the paper.

We understand the reviewer's request for a direct demonstration that Hyl1La processes miRNAs. We added sequencing results from Hyl1La IP samples that show no enrichment for other RNA molecules (Figure 5 – supplement 4) (see reply to essential revisions section 1). We also conducted an in vitro binding assay to show Hyl1La specific binding to pre-miRNA structures (Figure 6). In addition, our meta-analysis of miRNA profiles following knockdown of other miRNA biogenesis components shows clustering of Hyl1La morphants with Dicer1 and HEN1, which suggests they are involved in miRNA biogenesis in a similar manner. Together with our previous results and the new qPCR results for additional miRNAs we believe that we have a set of findings that support the notion that Hyl1La is involved in miRNA biogenesis in *Nematostella*. Regarding the suggestion to perform Hyl1La knockdown in adults, unfortunately it is currently technically impossible in *Nematostella* as there is no efficient method of delivery into adult *Nematostella* tissues.

2) A clear link between miRNA dysfunction and developmental phenotype.This would be a complementary way to try to demonstrate that the effect of HYL1 knockdown on development is through miRNAs. If the authors perform RNAseq over a timecourse of normal and HYL1 knockdown development, they could compare the two to find genes that are differentially expressed. These genes might be miRNA targets, and this could be associated using bioinformatic analysis to test whether the changes are consistent with the alterations in miRNAs observed upon HYL1 knockdown.

We thank the reviewer for the comment. In previous publications we show that we tried to test the expression of miRNA targets following Argonaute knockdowns and got a complex picture (Fridrich et al., 2020, Nature communications, 11(1): 1-12). In that set of experiments targets did not show expected behavior and other genes were affected. Notably, this is likely due to the fact that some miRNA targets in *Nematostella* are transcription factors such as Hox D and Six3/6 which were shown to play pivotal roles in *Nematostella* development (He et al. 2018 Science 361(6409): 1377-1380 Sinigaglia et al. 2013 PLOS Biology 11(2):e1001488.), hence regulating a complex network of downstream developmental effects. Therefore, the picture we get with such an experiment is not clear enough to show the direct effect of miRNA depletion on their targets. In addition, an experiment such as described will require enormous amounts of work we believe it is beyond the scope of this paper.

3) Another important thing to look at is to examine whether endogenous small RNAs are affected by the knockdown. The authors' model that this is a specific miRNA processing factor would indicate that other classes of small RNAs, including piRNAs and endosiRNAs, will be unaffected or minimally affected. The authors have already generated small RNA sequencing data so this would be straightforward to test.

Following this request, we extended the small RNA analysis to include piRNAs and siRNAs (please see reply to essential revisions section 2)

– Recommendations for improving the writing and presentation.I'm not sure that the authors should make the bold claim of common evolutionary history of plant and animal miRNA biogenesis in the introduction and abstract. I fully agree that this is an exciting implication, but the paper so far has not demonstrated this- if they were to do some of the experiments above perhaps this would be justified but otherwise it should be placed in the discussion but not elsewhere.

After carefully comparing our claims to the claims made by others in the miRNA field promoting the opposite view and the available evidence they use, we respectfully disagree regarding this point made by the reviewer. We believe that we are cautious enough in our claims about the homology and role of Hyl1La in *Nematostella* and that the presentation of our study is balanced in its current form.